**JCB** Journal of Cell Biology

# Stepwise assembly of α-hemolysin from intermediates to the mature pore in native erythrocytes

Arnab Chatterjee[1] , Anupam Roy[1] , Partho Pratim Das[1] , Debajyoti Chakraborty[1] , Bartika Ghoshal[2] , Siddharth Jhunjhunwala[2] , and Somnath Dutta[1]

**Alpha-hemolysin (α-HL) is a small pore-forming toxin secreted by pathogenic *Staphylococcus aureus*, inducing cell death process by forming pores in membrane. So far, detergents or artificial lipid environments have been utilized to characterize the toxin structure. The toxin-induced changes in the membrane, membrane remodeling after toxin treatment, and the role of the toxin during pore formation process remain ambiguous. Thus, understanding pore formation in the cellular environment, including the roles of the plasma membrane and lipid composition, is crucial for drug development. In this study, we captured step-by-step oligomerization of α-HL and membrane rupture of erythrocyte cells using confocal microscopy, cryo-EM imaging, and single-particle analysis. We resolved 3.1–3.8 Å resolution structures of pore, prepore, and immature pore conformations in cellular environment. Furthermore, mass spectrometry analysis demonstrated key erythrocyte lipid components interacting with α-HL. Our findings indicate that shorter or unsaturated lipid chains facilitate pore formation and the role of phosphatidylcholine with varying physical properties in modulating the toxin's activity.**

## Introduction

*Staphylococcus aureus* infections have become a global threat to human health due to the rapid emergence of methicillin-resistant *Staphylococcus aureus* and vancomycin-resistant strains. This is the most common bacterium that causes different infections ranging from skin and soft tissue infections to pneumonia, endocarditis, and sepsis (Lowy, 1998; Wertheim et al., 2005). The limited availability of antibiotics and the absence of approved vaccines make it extremely challenging to combat *S. aureus* (Chambers and DeLeo, 2009). To address this issue, it is crucial to study the pathophysiology of *S. aureus*. Pathogenic bacteria secrete various cytotoxic pore-forming toxins (PFTs) that primarily target host cell's plasma membrane, which is the protective barrier for the cell (Bischofberger et al., 2012; Dal Peraro and van der Goot, 2016; Valeva et al., 2006). Previous studies suggested that the level of PFTs directly correlates with the severity of the infection, and patients having a better clinical outcome have shown high anti-toxin (α-hemolysin [α-HL]) antibody levels (Ragle and Bubeck Wardenburg, 2009). Therefore, studying the mechanism of action of the toxins has become a crucial choice to develop a potent vaccine or immunotherapy. These PFTs assemble on the plasma membrane of the cells, and the transmembrane (TM) regions punch holes to create pores on the membrane, thereby

disrupting the cellular integrity (Kulma and Anderluh, 2021; Wilson et al., 2019). α-HL is one of the PFTs from *S. aureus* that oligomerizes in the presence of phosphatidylcholine (PC) in the membrane, and the β-hairpin structure forms the TM pore (Galdiero and Gouaux, 2004). α-HL comes under the β-PFT class of proteins, where the TM pore consists of β-strands (Song et al., 1996). Additionally, α-HL is a mono-component toxin, where a single polypeptide chain makes a protomer, and all the same protomers come close to each other to form an oligomer (Song et al., 1996). Therefore, the sequences corresponding to all β-strands forming the pore are identical. Stoddart et al. showed that sequentially deleting the TM β-barrel (up to 10 amino acids) region reduced lytic activity and prevented the oligomerization process (Stoddart et al., 2014). The reduction in lytic activity was monitored using the hemolytic assay, and the reduction in oligomerization was observed using SDS-PAGE analysis. Thus, understanding the barrel insertion/pore formation process could help to design potent therapeutic measures to prevent cell lysis.

The changes in the cellular morphology of leukemic cell lines were well established during the attack of large PFTs, where the cells shed microvesicles, helping in toxin clearance (Romero et al., 2017). However, α-HL is well known to lyse erythrocytes

---

[1]Molecular Biophysics Unit, Indian Institute of Science, Bengaluru, India;   [2]Department of Bioengineering, Indian Institute of Science, Bengaluru, India.

Correspondence to Somnath Dutta: somnath@iisc.ac.in.

and is cytotoxic against human leukemia-60 (HL-60) cell lines, but the role of TM β-barrel during this lysis process is unclear (Cassidy and Harshman, 1973; Tran et al., 2020; Tsuiji et al., 2019). Additionally, α-HL was reported to induce necrotic cell death in Jurkat cells and apoptosis in human platelets (Essmann et al., 2003; Jahn et al., 2022). Therefore, it is important to understand the toxin-mediated cell lysis or cell death process and associated toxin conformations during the hemolysis event.

Previously, many groups have solved the crystal structures of detergent-solubilized and lipid-exchanged heptameric pore of α-HL. The researchers have predicted a few amino acids in the rim region responsible for lipid interaction (Galdiero and Gouaux, 2004; Tanaka et al., 2011). However, these studies have not illustrated the membrane interaction regions of the toxin and other possible intermediate species. Therefore, structural studies of PFTs in the presence of host cells were required to understand these events in physiological conditions. Although there have been a few reports on toxin-mediated alterations in host cell membrane morphology, none have specifically investigated these changes at atomic or near-atomic resolution during the action of small PFTs (Essmann et al., 2003; Fiaschi et al., 2016; Krones et al., 2021). Such high-resolution structural insights are critical for understanding how cellular morphology is altered upon toxin attack. These changes could provide us with a better understanding of the roles of distinct lipid components in toxin interactions, as well as the involvement of membrane compositions during the membrane repair process. Therefore, for the rational design of novel inhibitors, it is essential not only to understand the mechanisms of pore formation and toxin oligomerization but also to elucidate how the target cell membrane's structural and physicochemical properties are modulated during toxin exposure and infection. Several studies have attempted to capture prepore states of PFTs using mutants, lipid modulation, or low-temperature conditions. For example, cryo-electron microscopy (cryo-EM) analysis of actinoporin fragaceatoxin C in the presence of sphingomyelin-rich membranes revealed a 30 Å low-resolution octameric prepore structure where the TM region was missing (Morante et al., 2016). Prepore states were also observed in insecticidal membrane attack complex/perforin proteins (Mpf2Ba1) in gut fluidic conditions, perfringolysin O at low-temperature conditions, and perforin-2 in the presence of liposomes, as well as in Vibrio α-HL under low $Ca^{2+}$ conditions (Chiu et al., 2023; Marini et al., 2023; Shepard et al., 2000; Wade et al., 2019; Yu et al., 2022). While the structure determination of the detergent-solubilized inactive mutant of α-HL resulted in the formation of a pore conformation (Sugawara et al., 2015), the detergent became an unusual choice to capture the prepore states. Cryo-EM of Vibrio cholerae cytolysin provided low-resolution snapshots of prepores of varying barrel lengths in the presence of liposomes (Sengupta et al., 2021). Similarly, high-resolution pore and prepore conformations of heptameric α-HL were resolved in the presence of PC-containing liposomes (Chatterjee et al., 2025b). Overall, the structural basis of prepore states in native cellular conditions is still elusive.

Therefore, in this current study we wanted to capture the different possible prepore species that exist in the cellular environments. Thus, we incubated the monomeric toxin with the target host cell, erythrocyte, and performed a structural and functional analysis of the oligomers of α-HL in the pre- and posthemolytic stages. Our cryo-EM–based structural analysis revealed the presence of many intermediate species, such as arc-like oligomers, different heptameric conformations, and octamers in the cellular condition. In addition, we evaluated the morphology and membrane-related changes occurring due to toxin attack, which included membrane protrusion followed by a necrotic cell death. The role of different chain lengths and saturation of lipids during toxin binding and the key interaction amino acids was also identified. This assisted us in interpreting the lipid–protein interface during its interaction with the host plasma membrane. Moreover, this study provided a detailed overview of structural–functional analysis of different possible toxin conformations in a cellular environment and membrane-associated changes during the pore formation process.

## Results

### α-HL–mediated effect on HL-60 cells

Our target was to characterize the effect of α-HL on neutrophil-like HL-60 cell, which is a crucial component of the host defense mechanism. HL-60 cell lines were considered for the study, as these cells have a phenotype similar to neutrophil cells, and are a common cell line to use as a substitute for neutrophil cells (Carrigan et al., 2007; Hauert et al., 2002; Woo et al., 2003). To understand the crosstalk between α-HL and the plasma membrane during interaction with HL-60 cells, the recombinant toxin was purified in a two-step purification process. Ni-NTA affinity–purified protein was used, followed by size exclusion–based purification. The SEC profile depicted a sharp peak at 17 ml, corresponding to the molecular weight of the monomeric toxin (Fig. S1 A). The SDS-PAGE of the eluted protein also showed a prominent band at 33 kDa, confirming the purity and size of the protein (Fig. S1 B). To understand oligomer-induced membrane damage by the α-HL toxin, a DNA-binding dye, 4′,6-diamidino-2-phenylindole (DAPI), was introduced along with the toxin (100 nM) in the growth medium of HL-60, which resulted in labeling of the nucleus (Fig. 1 A). HL-60 cells initially stained three lobes of the nucleus because of DAPI binding to the nucleus; however, after 5 min, the intensity of DAPI was significantly reduced in HL-60 cells (Fig. 1 A). This might be possible that due to membrane rupture, the nucleus of the HL-60 cells was disrupted. Additionally, we observed protrusions of small vesicles from the plasma membrane, confirming the membrane rupture and cell death (Fig. 1 B; and Videos 1 and 2).

To determine the cell death pathway induced by the purified toxin in a quantitative manner, flow cytometry of toxin-treated cells was performed using propidium iodide (PI) fluorophore. At 10 nM toxin concentrations, HL-60 cells survived, whereas following incubation with higher concentrations (100 nM and 1 μM), cell populations shifted toward the necrosis-mediated cell death pathway, as evident by the increase in PI-positive cells (Fig. 1 C). This necrosis-mediated cell death process happened in a concentration-dependent manner and damaged the HL-60 cell membrane. So far, we evaluated the morphological changes in

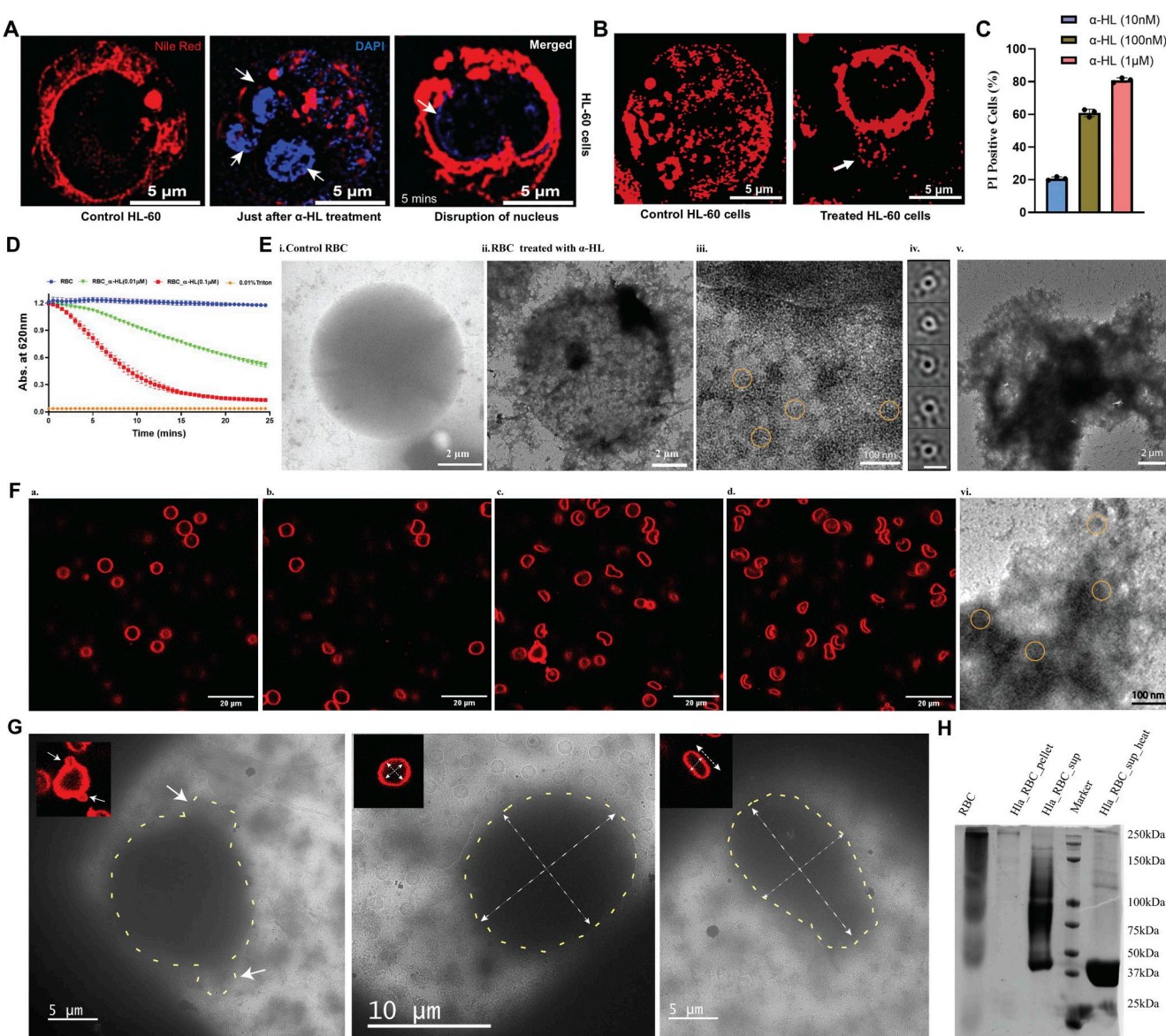

Figure 1.  **Effect on HL-60 cells and RBCs after α-HL. (A)** Intensity of the nucleus (in blue color) staining with DAPI dye ($\lambda_{excitation}$= 405 nm, $\lambda_{emission}$= 480–510 nm) increased for the first couple of minutes, where the three nuclear lobes (white arrowhead) became clearly visible due to the uptake of the dye from the outside medium. After some time (5 min), the intensity of the nucleus started decreasing, and three distinct lobes of the nucleus started disappearing. Nile red intensity ($\lambda_{excitation}$= 540 nm, $\lambda_{emission}$= 565–610 nm) corresponding to the membrane (in red color) started decreasing rapidly just after toxin treatment at a 100 nM toxin concentration. **(B)** Toxin treatment showed membrane protrusion (in red color) from HL-60 cells at 100 nM toxin concentration. **(C)** Flow cytometry analysis of 10 nM, 100 nM, and 1 µM toxin-treated HL-60 cells stained with PI. Compared with the untreated cells, the toxin-treated cells started becoming necrotic, as evident by the increase in PI-positive cells. Data are shown as the mean ± SD of triplicate measurements ($n$ = 3 biological triplicates). **(D)** Concentration-dependent hemolysis of rabbit erythrocytes (10%) was monitored after adding different concentrations of purified monomeric α-HL at 37°C. **(E)** Different stages of RBC damage were captured using NS-TEM after adding the toxin: i. untreated control RBC; ii. damaged membrane of RBCs after α-HL addition; iii. zoomed view of the membrane area showed circular-shaped particles (yellow encircled); iv. 2D class averages of the particles confirmed the circular shape of the toxin (scale bar: 10 nm); v. lysed cell after completion of hemolysis; and vi. a zoomed view of the lysed-membrane area showed circular-shaped particles (yellow encircled). **(F)** Toxin-incubated RBC samples were imaged under a confocal microscope at different time points: **(a)** just after toxin addition; **(b)** starting point of membrane damage 2 mins post-toxin addition; **(c)** lysis initiation state after 5 mins; and **(d)** mostly lysed cells after 10 min. The cell membranes (in red color) were stained using Nile red dye ($\lambda_{excitation}$= 540 nm, $\lambda_{emission}$= 565–610 nm). **(G)** Correlation of toxin-mediated damage in RBCs using high-resolution cryo-EM imaging with the confocal data (shown in top left). The protruded portions' size from cells under confocal and cryo-EM in the left side panel was around 1 µm. The ratio of width to length of cells under confocal and cryo-EM was 1.2 in the middle panel. The ratio of width to length of cells under confocal and cryo-EM was 0.74 in the right side panel. **(H)** SDS-PAGE of RBC treated with α-HL (from left side to right side, control RBC, RBC pellet after toxin incubation, supernatant from the toxin-incubated sample, protein marker, supernatant sample after heating). RBCs, red blood cells. Source data are available for this figure: SourceData F1.

the membrane during toxin attack using fluorescence microscopy and characterized the cell death pathways. However, we were unable to visualize the conformational states of the toxin during its interaction with the membrane. Thus, we performed cryo-EM–based structural analysis of toxin-treated HL-60 cells to visualize the morphology of α-HL in the cellular condition. The cryo-EM micrographs depicted the presence of circular toxins on the cell membrane. However, we failed to obtain high-resolution 2D class averages from this dataset (Fig. S1 C). One of the possibilities could be the rupturing of HL-60 cells during their interaction with the toxin, which resulted in the release of intracellular material to the outside medium. Therefore, we considered a cell type that is also affected by the toxin for structural analysis but has fewer genetic components. Additionally, to make the structural studies more physiologically relevant, we wanted to use cells isolated from animals, instead of using cultured cells.

### α-HL–mediated effect on erythrocytes during hemolysis

The primary target cell of α-HL is erythrocyte cells (DuMont and Torres, 2014). Therefore, we considered using rabbit red blood cells for the oligomerization and further structural analysis of the toxin, as well as to study the morphological changes of the outer membrane of the RBC. The RBCs were washed thoroughly to remove the cell debris before incubating the toxins to improve the sample quality for cryo-EM. To check the cytotoxic effect imparted by the α-HL, we performed a hemolytic assay, where a 100 nM toxin concentration was found to lyse all the cells in 10 min (Fig. 1 D). Further, we checked the changes in the cellular morphology and plasma membrane during this hemolysis process. The toxin-treated RBCs were imaged under TEM at the initial time point (prehemolytic) and at the end of hemolysis (posthemolytic). In the prehemolytic condition, leakage from the cells and membrane roughness were clearly visible (Fig. 1 E). The enlarged micrograph showed the presence of circular particles in the membrane periphery. The 2D class averages of those particles resembled the ringlike appearance of the oligomeric α-HL (Fig. 1 E). However, in the posthemolytic condition, complete lysis of the erythrocytes was also observed (Fig. 1 E). In addition, to understand morphological changes over the entire hemolysis period, time-lapse imaging was performed using fluorescence microscopy, where, over time, the morphological changes of RBCs with α-HL were monitored. In the initial time period, postincubation of α-HL, membrane protrusion, and shape changes in some of the cells were observed (Fig. 1 F, Fig. S1 D, and Video 3). However, the morphological changes became distinct as the hemolysis progressed. In the end, the shape of the entire cell population was found to be distorted (Fig. 1 F and Fig. S1 D). The protrusion of the plasma membrane was further confirmed using high-resolution imaging (Fig. S1, E and F; and Videos 4, 5, and 6). The TEM analysis of the entire cell with toxin showed a lot of cellular debris that can significantly impact cryo-EM–based structural analysis. However, to visualize the overall cellular morphology by cryo-EM, we observed the cells treated with toxins at low magnifications. The NS-TEM imaging of toxin-treated and untreated cells showed a clear difference in morphology (Fig. S2, A and B). The bilayer appeared smooth in

control RBCs, where the membrane bilayer was visible. However, the membrane became rough and uneven in toxin-treated cells (Fig. S2, A and B). The low-magnification cryo-EM images of whole RBCs showed a damaged and blurred appearance in the case of the posthemolytic cell as compared to the prehemolytic cell, and the control RBC image (Fig. 1 G and Fig. S2, C–E). A correlative imaging approach was performed using cryo-EM to validate our low-resolution fluorescence microscopy–based RBC lysis images. Therefore, we prepared parallel samples for both confocal microscopy and cryo-EM using identical cell and toxin concentrations under the same experimental conditions. The membrane protrusion and membrane distortion were observed in the cryo-EM micrograph (Fig. 1 G). Thus, different stages of RBC membrane damage induced by the toxin were captured using fluorescence microscopy, as well as cryo-EM–based approaches. Further structural analysis was performed to understand the different states of α-HL generated after hemolysis.

### Structural analysis of different states of α-HL oligomers in the posthemolytic condition

We wanted to isolate oligomeric toxins from the posthemolysis solution for structural analysis. Thus, centrifugation of the toxin–RBC solution was performed at 18,328 × g for 1 min to separate ruptured RBCs, cell debris, and toxin-induced cell supernatant. The SDS-PAGE analysis of the pellet and supernatant fractions showed that the protein was mostly present in the supernatant (Fig. 1 H and Fig. S3 A). The TEM analysis of the supernatant revealed the presence of the oligomeric toxins on ruptured membranes, as well as free oligomers (Fig. S3 B). The TEM images indicated that the supernatant was suitable for visualization of RBC membrane-embedded oligomeric toxins, which were homogeneously distributed in the grid. Additionally, SDS-PAGE of the supernatant showed a clear band just above 37 kDa, which also indicated that it is a mixture of toxins embedded in ruptured RBC membrane (Fig. 1 H). This experiment suggested proceeding with the supernatant fraction for cryo-EM sample preparation to evaluate different states of the toxin. The cryo-EM micrograph showed the presence of circular toxin and plasma membrane containing the toxin. The 2D class averages of the particles confirmed the ring-shaped heptameric toxin and mushroom-shaped side views of the toxin (Fig. 2, A and B; Fig. S2 F; and Fig. S3 C). The 3D classification from these particles resulted in different heptameric conformations of the α-HL oligomer (Fig. S3 C). Apart from the pore form, where the β-barrel was completed, another conformation appeared where the β-barrel was absent. The conformations without a complete β-barrel were termed as a prepore state. Here, we resolved the pore conformation at a resolution of 3.1 Å (Fig. 2 C). This is the first high-resolution structure of any small β-PFTs in the cellular environment. At a high threshold, the 14 TM β-strands were resolved clearly. This high-resolution map was considered for atomic model building, and the side chains of the amino acids fit appropriately in the EM density (Fig. S3 C), whereas at a lower threshold, a clear extra density was visible surrounding the TM segment and lower rim domain (Fig. 2 C; and Fig. S3, C and D). We speculated that density was coming from the membrane components and was found to be nicely decorating the

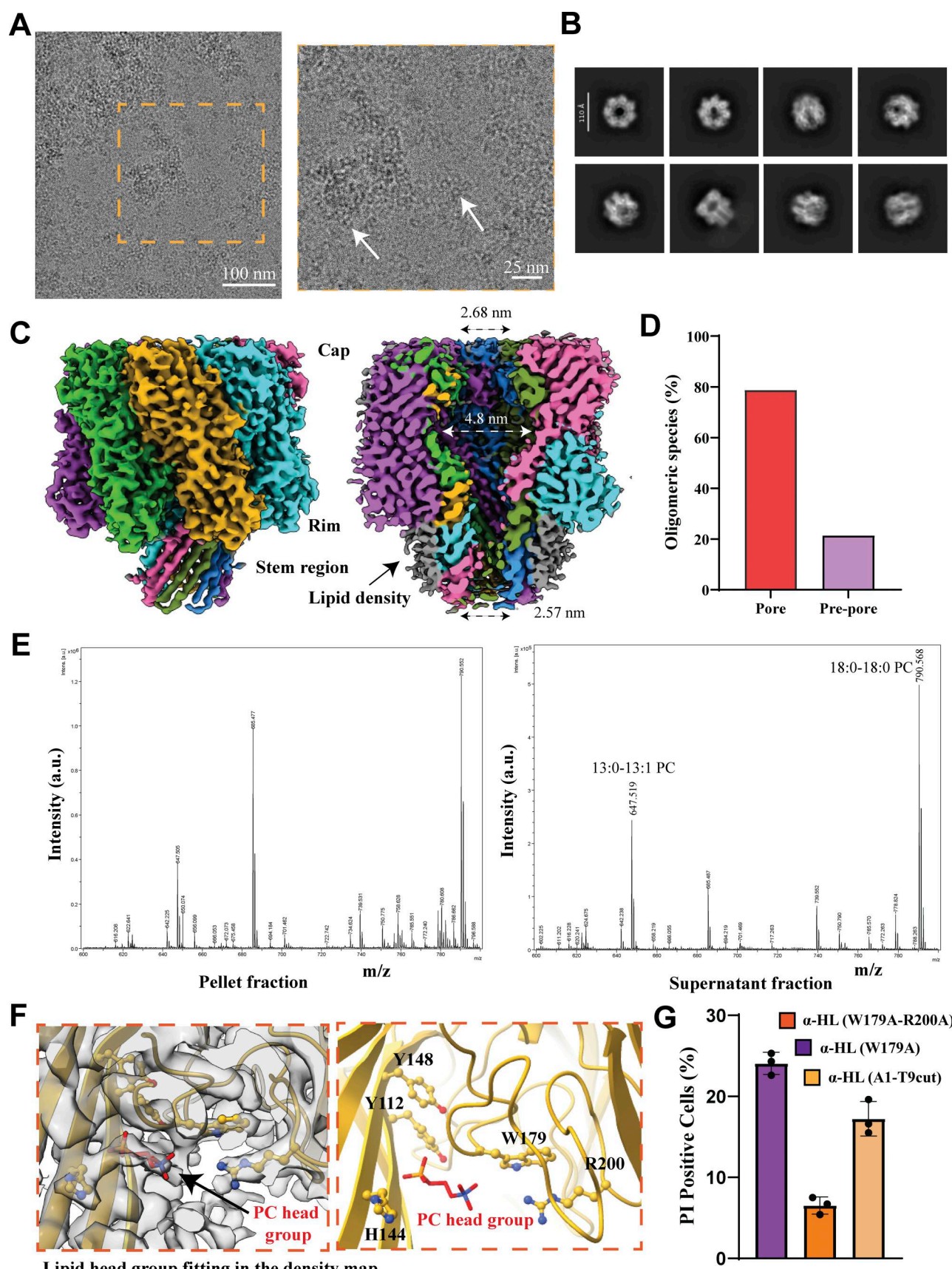

Figure 2. **Structural analysis of α-HL heptamer in the presence of RBC environment. (A)** Cryo-EM micrograph of α-HL oligomers in the RBC environment at a magnification of 54 k× and zoomed view showed circular toxins (white arrowhead). **(B)** 2D class averages showed disk-shaped top views of toxins and

mushroom-shaped side views. **(C)** Cryo-EM maps of α-HL heptameric pore structure at 3.1 Å resolution; each protomer is shown in different colors (from left, lime green, golden rod, cyan, hot pink, olive drab, dodger blue, medium orchid). The cross-section of the map showed a pore channel and extra density (gray color) corresponding to the membrane surrounding the barrel and the lower part of the rim domain. **(D)** Particle distribution after 3D classification shown in a bar plot (heptameric pore species in red color and heptameric prepore in purple color). **(E)** Mass spectrometric analysis of lipids associated with the toxin. Left panel showing the spectrum of lipids present in the pellet fraction, and the right panel showing the spectrum of lipids present in the supernatant fraction. **(F)** PC head group fitting in the EM density map (gray color). Seven PC head groups (red color) were fitted symmetrically at the topmost density in between the rim and stem domain. The interacting amino acids (golden color) from the rim and stem domains with the lipid are Y112, H144, Y148, W179, and R200. **(G)** Flow cytometry analysis of PI-positive HL-60 cells after incubation with mutant toxins (100 nM concentration). Data are shown as the mean ± SD of triplicate measurements (*n* = 3 biological triplicates).

membrane-binding sites (rim and stem domains) of the toxin (Fig. S3, C and D). Interestingly, a small population of prepore species also existed in this condition, where β-barrel was not formed (Fig. 2 D and Fig. S3 C). The less particle number of prepore species prevented us from obtaining a high-resolution structure. Additionally, we hypothesized that these prepore conformations are highly flexible and unstable, as they might behave as a membrane-anchored protein, where the TM segment was mixed with the plasma membrane. Therefore, solving the structures of these states was difficult due to the flexible nature of the prepore conformations. Additionally, as PC is the oligomerization agent for α-HL, we hypothesized that different chain lengths of PC should get attached to the toxins, which could modulate the toxin binding. Therefore, we performed mass spectrometry of these toxins only to identify lipid moieties attached to the toxin molecules. The mass spectrometry (matrix-assisted laser desorption/ionization [MALDI]) analysis showed the presence of different lipid components bound with the toxin's β-barrel and rim domain, which was further used for structural analysis (Fig. 2 E). The MALDI data suggested the peaks having high abundance 647.52 g/mol and 790.56 g/mol corresponding to molecular weight 13:0–13:1 PC and 18:0–18:0 PC, respectively, which suggested that toxin preferred binding to an area where the membrane was flexible due to the presence of both unsaturated and saturated lipids with different chain lengths. These results provided an idea of the involvement of different lipid chain lengths and unsaturation that could modulate the toxin binding. Therefore, it was crucial to characterize the lipid–protein interface to understand the amino acids involved in lipid interaction. Hence, we tried to fit the PC head group into the lipid density between the stem and rim domains. The fitted lipid densities showed the nearby interacting amino acids, Y112, H144, W179, and R200 (Fig. 2 F). We could also observe that amino acids (Y112, H144) from the stem domain are involved, along with previously known lipid-interacting residues from the rim domain (Galdiero and Gouaux, 2004). The hemolytically inactive mutants of these lipid-binding residues (W179 and R200) were found to show minimal cytotoxic effect on HL-60 cells. The double mutant (W179A-R200A) also showed less cell killing compared with the single mutant (W179A) (Fig. 2 G). Furthermore, we performed a detailed comparison of the pore structure solved from RBCs with detergent-solubilized and liposome-based stabilized pore structures (Fig. 3, A–D). The root mean square deviation (RMSD) of pore structure from RBCs with 2-methyl-2,4-pentanediol (MPD)–solubilized pore (Protein Data Bank [PDB] ID: 3ANZ), pore crystal structure with PC head group (PDB ID: 6U49), 10:0 PC liposome–derived pore structure

(PDB ID: 9KRF), and detergent-solubilized pore structure (PDB ID: 7AHL) is 0.548, 0.560, 0.152, and 0.532 Å, respectively (Fig. 3, A–D). These comparative studies demonstrate that the α-HL toxin's overall structures in the cellular environment, particularly the cap region, are highly similar to the detergent-solubilized crystal structure of α-HL. However, minor differences were observed in the rim and stem regions, which directly interact with the lipid membrane. Notably, a shift was detected in the rim domain loops (at residues S262 and G180, displaced by 2.4 and ~2 Å, respectively) when our structure was compared with the crystal structures (Fig. S3 E). Thus, we were able to identify minor structural differences between the detergent-solubilized crystal structure and the lipid-binding α-HL structure reported in this manuscript. Additionally, the lipid head group–binding site we calculated from our structure matches with crystal structure with MPD and lipid head group (Fig. 3, A and C). Moreover, we were interested in capturing intermediate oligomeric species of wild-type α-HL in cellular condition that was completely unknown until now.

## Appearance of different α-HL oligomers in the prehemolytic condition

We considered two methods that could help us to trap the transient states of this toxin in larger quantities and in a relatively stable form. Firstly, we could rigidify the plasma membrane in such a way that toxins were unable to insert any prestem into the membrane during the β-barrel formation step. Secondly, we could isolate the toxin molecule in prehemolytic conditions before completing the entire barrel formation. Therefore, to validate our hypothesis, we either need to slightly lower the reaction temperature below room temperature to slow down the process or drastically reduce the reaction time (by mixing the toxins with RBCs and immediately freezing the sample) so that the toxin protomers do not have enough time to oligomerize. Thus, in order to validate our hypothesis, we checked whether incubation of the toxin at a reduced temperature had any effect on RBC lysis, and the hemolytic assay was performed at 10°C. The hemolytic assay of rabbit erythrocytes at different toxin concentrations showed an increase in the lag phase of the hemolysis at 10°C, suggesting a significant reduction in the membrane perforation process by the toxin (Fig. 4 A). To further confirm the lower temperature-mediated increase in the compactness of the plasma membrane, we followed a solvatochromic dye-based approach, where a higher emission band in the 525- to 565-nm range was observed in the RBC membrane kept at 4°C as compared to the RBC kept at 37°C for 1 h (Fig. 4 B). This indicated that the compact packing of the lipid components

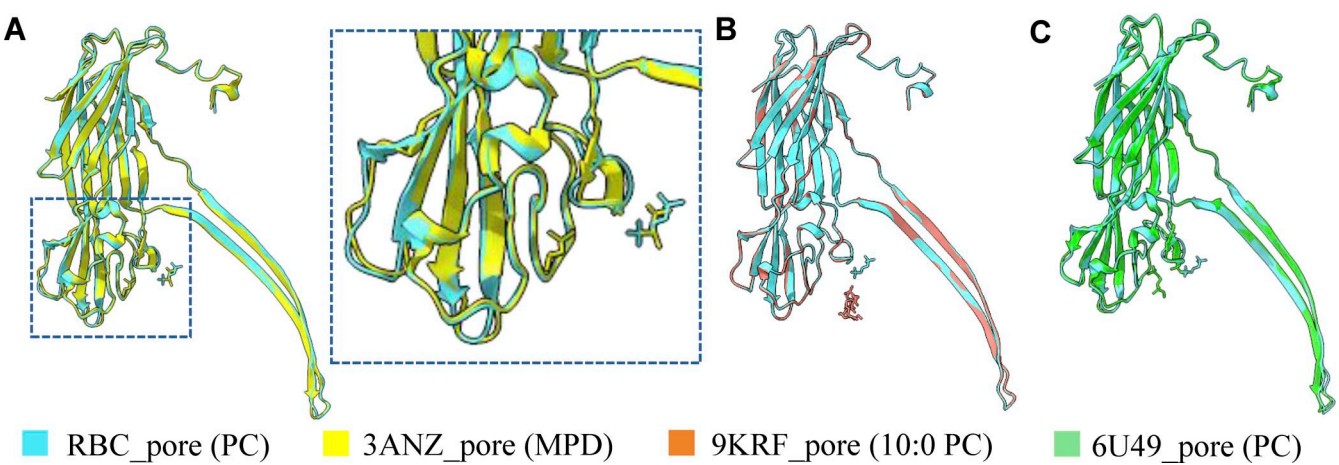

Figure 3. **Comparison of lipid-binding sites in different detergent-solubilized and liposome-based stabilized pore structures. (A)** Comparison of pore structure in the presence of RBCs (cyan) with pore in the presence of MPD (yellow). The zoomed view shows the location of the lipid head group and MPD density. **(B)** Comparison of pore structure solved in the presence of RBCs (cyan) with pore structure solved in the presence of 10:0 PC liposomes (orange). The zoomed view shows the location of the lipid head group and 10:0 PC density. **(C)** Comparison of pore structure solved in the presence of RBCs (cyan) with pore structure solved in the presence of $C_{14}$ PC (green).

of the RBC plasma membrane led to the formation of a relatively ordered membrane bilayer. We also wanted to confirm whether any change in the secondary structure of the toxin happened during this cold condition. As expected, there were no changes observed in the secondary structure as observed from the circular dichroism analysis (Fig. 4 C). Hence, the changes in the protein structure would be due to lipid–protein interaction. These biophysical data convinced us to proceed with structural analysis. The toxin with the RBCs was immediately centrifuged at 4°C at 18,328 × g for 1 min, just before plunge freezing, and the supernatant fraction was considered for cryo-EM analysis (Fig. S3 A). The cryo-EM micrograph of the prehemolytic sample showed fragmented plasma membranes along the ringlike circular toxins (Fig. 4 D and Fig. S2 G). A careful particle picking followed by 2D classification provided toxins of different geometric shapes. Surprisingly, along with the heptameric form, arc-like oligomers and octameric forms were also observed (Fig. 4, D and E). Furthermore, a detailed analysis of the 2D classification revealed that all arc-like structures consist of distinct subunit arrangements. From these 2D class averages, we were able to identify partial dimers, tetramers, pentamers, and hexamers (Fig. 4 E and Fig. S6 A). This unique observation highlights how single-particle cryo-EM was effectively employed to capture the stepwise, subunit-by-subunit oligomerization of α-HL on a native cellular membrane. This provided an excellent opportunity to capture a rare event where we were able to visualize the intermediate states of toxin assembly. The shape of the arc-like species clearly depicted an intermediate form during the oligomer formation. Additionally, octameric oligomers were also observed, which was fascinating (Fig. 4 D and Fig. S6 D). This octameric structure also appeared to have a ringlike shape but was not completely circular, unlike the heptameric species. We could see such geometric species for the first time due to the reduced incubation temperature and the shorter incubation time period of the toxin with RBCs, which strongly supports our hypothesis. The highly detailed 2D averages

encouraged us to perform single-particle cryo-EM analysis to determine high-resolution structural analysis.

## Structural analysis of the different oligomeric species

So far, our cryo-EM results have demonstrated various intermediate species, and we further proceed with the structural analysis to understand the lipid-based stabilization of each species. The separated heptameric populations underwent rigorous 3D classification to separate different conformational heterogeneity. Based on the length of the barrel, we classified different conformations into different prepore states (Fig. S4). We refined distinct conformational states to get a high-resolution structure. Interestingly, there was no conformation with a full barrel state in this condition (Fig. S4). From the side views of these prepore states, it was clear that the barrel was crossing the rim domain only in the prepore state IV (Fig. S4). The prepore states Ia and Ib did not consist of a barrel (Fig. S4 and Fig. S5 A). This is the first prepore structure without the stem region, but as these are very unstable species, we could not get a high-resolution structure to generate any atomic model corresponding to these structures. In the prepore state II, the first few amino acid densities in the upper non-TM segment of the barrel were visible at a higher threshold (Fig. 5, A and B). At a low threshold, the pore vestibule was completely blocked, suggesting that the rest of the barrel was in a mixed condition with the plasma membrane (Fig. S5 B). Hence, the prepore state I and prepore state II were possibly hemolytically less potent states, as the TM segment could not pass the bilayer. The prepore state III was the most interesting state. At a high threshold, the densities corresponded to the first 6–7 amino acids of the upper part of the barrel were resolved (Fig. 5, C and D; and Fig. S5 C). But a slight decrease in the threshold value resulted in the visualization of an extra density corresponding to TM β-strands spread out to the outside environment while connected to the upper barrel part (Fig. 5, C and D; and Fig. S5 C). It is possible that the bending of the barrel starts from the point where the TM segment hits the

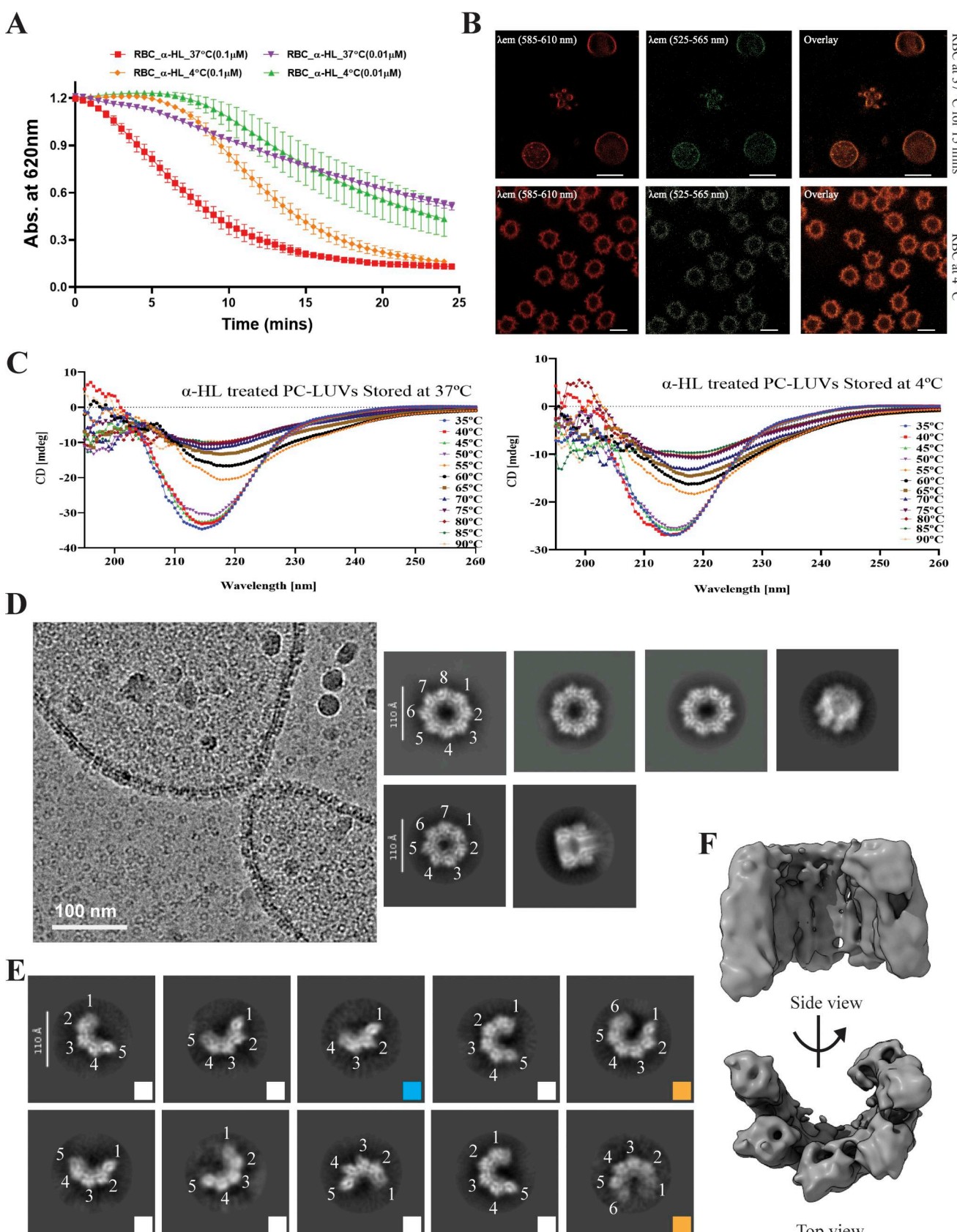

**Figure 4.** **Conformational and functional analysis of α-HL oligomers in an instant hemolysis environment. (A)** Compromised hemolysis of RBC at a lower temperature (10°C) with different toxin concentrations as compared to hemolysis at 37°C. **(B)** Membrane rigidity was measured between RBCs kept at 37°C (for 15 min) and 4°C (for 1 h) using a solvatochromic dye Nile red (emission range between 585 and 610 nm showed in red color, and 525 and 565 nm

showed in green color). **(C)** CD analysis of toxin at different temperatures with or without oligomerizing agents (POPC-LUVs). **(D)** Cryo-EM micrograph showing toxin particles, and 2D class averages showing heptameric and octameric states. **(E)** 2D class averages of different sizes of arcs are highlighted in different square boxes (blue—4-mer or tetramer; white—5-mer or pentamer; and orange—6-mer or hexamer). **(F)** 3D reconstruction (top and side views, in gray color) of the pentameric arc-like intermediate species. CD, circular dichroism.

membrane. It can be one of the intermediate states during the channel formation, where the rest of the barrel region tries to establish proper interaction with the lipids during the pore formation and form a stable symmetric channel. Another possibility is that the rigid membrane did not allow the barrel to penetrate the bilayer and forced the TM region to be distorted. As plasma membranes also form microdomains, this could be a transient species that could exist during oligomerization of the toxin in the microdomain or rigid bilayer region. Similar to the previous prepore, the vestibule region of the prepore state III was also blocked by the lipid density, indicating that the barrel could not cross the membrane bilayer (Fig. S5 C), whereas the prepore state IV had a stable stem region except for the lower part of the TM segment (Fig. 5, E and F; and Fig. S5 D). Compared with other prepore states, the barrel was the longest in this conformation. It was very important to understand whether this state could perforate the entire bilayer. However, when the map was analyzed at a lower threshold, the pore vestibule was still blocked by the lipid densities (Fig. S5 D). This result indicated that this barrel length was not sufficient to span the entire membrane. However, the extension of the barrel inside the cell membrane in the prepore states III and IV suggested that these states can make a significant impact on the hemolysis process. The low threshold maps of pore structure revealed that the barrel region was not blocked by lipids, suggesting that the obtained mature pore species was hemolytically active (Fig. S5 E).

Additionally, we calculated the distribution of different species in a prehemolytic condition. In this condition, the heptamers were the predominant species (73%), whereas 23% of octamers and a small population of arc-like species of 4% (Fig. 5 G) were observed. On the other hand, among the heptameric species, the count of the prepore state Ia and prepore state IV was twice that of the prepore state Ib and prepore state III (Fig. 5 H). Until now, we understood that the prepore state I and prepore state II acted as a membrane-anchored species, therefore making it the most unstable species; thus, it was an extremely challenging task to resolve the high-resolution structures of the prepore state I and II. Moreover, the stabilization of species directly correlates with the stability of the membrane-inserted stem region. Instead of a small population of arc-like species, we made an attempt to solve the structure to mainly understand the position of the stem region (Fig. 4 F; and Fig. S6, A and B). Population distribution of the arc-like species showed that the 5-mer species were high in number as compared to the 4-mer or 6-mer (Fig. 5 I). Therefore, we refined 5-mer species without applying any symmetry to validate the hypothesis. From the refined low-resolution map, the boundaries of each protomer were clear, but the secondary structures were not resolved. Upon fitting five protomers from a pore structure, we could not see any extra density where the prestem could be accommodated (Fig. S6 B). This result suggested that the

detachment of the prestem region happened during the early stage of oligomer formation itself in the presence of membranes. Furthermore, our structural analysis showed that the presence of the adjacent protomer was very crucial in order to stabilize the N-terminal region by interacting with the upper cap region of the neighboring protomer (Fig. S6 C). Therefore, the N-terminal cut (A1–T9) mutant showed very little cytotoxic effect on HL-60 cells, where only 18% of the cell population died after 100 nM toxin treatment for 90 min (Fig. 2 G). This study matches with a previous study, where this mutant was demonstrated to be hemolytically inactive, as this stretch was responsible for opening the prestem domain of the adjacent protomer (Jayasinghe et al., 2006). However, our study is direct structural evidence that N-terminal region interacts with upper cap domain of the same and adjacent protomer to release the prestem domain. If adjacent protomer is not absent, the N-terminal region would be unstable and unable to detach the prestem. Then, we were curious to understand the geometry and location of the stem domain in the octameric population. The best class from the 3D classification was taken forward for refinement (Fig. S6 D). As the shape of this octamer is not completely circular but rather an oval shape, symmetry was not applied during refinement. The octameric structure was resolved at a low resolution of 5.9 Å (Fig. S6 D). When the threshold was decreased, the vestibule was entirely blocked by an extra density. Similar to heptameric prepore species, the bottom portion of the octamers consisted of the flexible stem region and lipids (Fig. S6 D). Thus, we were speculating that both arc-like and octameric species existed as membrane-anchored species, which were also valid for the prepore state II. Also, the disorder N terminus needs to be stabilized for oligomerization, and the prestem should be detached for pore formation. These two events are spontaneous processes and should happen sequentially to produce active pores.

### Structural comparison of different heptameric conformations

Furthermore, we wanted to characterize the conformational changes in the heptameric prepore states to understand the barrel formation procedure. The high-resolution structures from the prepore state II to the state IV enabled the calculation of the corresponding atomic models (Fig. 6, A–D). Moreover, these atomic models helped to compare these states with each other and complete pore structure to understand the changes. There were no significant conformational differences except the missing densities in the lower TM region and the cytosolic loop in the prepore state IV when compared to the pore structure (Fig. 5, A–F). The analogy between the prepore state II and IV showed the missing densities of the TM segment, where the visible residues from the non-TM segments were D108, T109, K110, Y148, V149, and Q150 (Fig. 5 J and Fig. 6 F). In this case, we were also able to see a small upward shift of the loop present in the prepore state II as compared to the state IV. The comparison

Figure 5. **Structural comparison between heptameric states consisting of different lengths of β-barrel. (A)** Cryo-EM map of the prepore state II, density corresponds to one protomer shown in golden rod color, and the rest of the protomers in gray color. **(B)** Fitted atomic model of one protomer. **(C)** Cryo-EM map

of the prepore state III, one protomer is shown in blue. **(D)** Fitted atomic model of one protomer of the prepore state III. **(E)** Cryo-EM map of the prepore state IV, where density corresponding to one protomer is shown in deep pink color and the remaining six protomers in gray. **(F)** Fitted atomic model. **(G)** Population distribution of different oligomeric states. **(H)** Population distribution of different heptameric states in the prehemolytic condition. **(I)** Population distribution of different arc-like states. **(J)** Comparison between the prepore state II (golden) vs state IV (deep pink). The D108, T109, K110, Y148, V149, and Q150 residues are shown in deep green color. The minor upward shift on some loops is highlighted using a black arrow. **(K)** Comparison between the prepore state III (blue) vs state IV (deep pink). **(L)** Comparison between the prepore state IV (deep pink) and pore state (green).

of the prepore state III with the state IV and pore structure showed a visible change in the stem region, where a bending in the stem region was observed (Fig. 5 K). Two hydrophilic residues (Y112 and Y148) were marked at the bending starting point, which might act as initial lipid interaction sites, as well as maintain the interprotomeric contact to form the channel (Fig. 6 E). Additionally, an upward movement in the loops in the lower rim domain of the prepore state III was observed as compared to the pore and prepore state IV (Fig. 5 L). Previous reports suggested that this loop region contains lipid-binding residues, which further indicated that the slight conformational changes in the loops could be due to interaction with the plasma membrane (Galdiero and Gouaux, 2004). Thus, this structural analysis suggested that both rim and stem regions participated in lipid interaction to form an oligomer and subsequent barrel formation that led to the conformational changes in those regions in the prepore state.

## Discussion

α-HL is the most dominant PFT produced by *S. aureus* strains and has been shown to play a crucial role in bacterial pathogenesis. Since the discovery of neutralization of infection by anti-α-HL antibodies, studying the cytotoxicity and membrane lysis during pore formation has become critical (Ragle and Bubeck Wardenburg, 2009). Moreover, the structural details of plasma membrane–mediated oligomerization and the pore formation process of small β-PFTs remain elusive. However, PFTs have evolved to penetrate the cell membrane via different methods and damage the cell membrane. This includes a multistep oligomerization process of PFTs in the presence of different lipid components, membrane receptors, or fluidity of the membrane (Rojko and Anderluh, 2015). Modulating the biomembranes and generating inactive mutants have been popular choices to identify prepore or intermediate oligomeric species to understand the oligomerization mechanism. Previous study demonstrated that actinoporin fragaceatoxin C binds and forms pores in the sphingomyelin-rich membrane region, where DOPC-rich liposomes were used without adding sphingomyelin to obtain prepore states (Morante et al., 2016). Unlike FragC, α-HL does not have affinity to the ordered region (sphingomyelin-rich) of the cell membrane. However, structural analysis of α-HL in the presence of sphingomyelin-containing PC-containing liposomes revealed heptameric prepore species, whereas the PC-containing liposomes without sphingomyelin generated heptameric pore conformation (Chatterjee et al., 2025b). This study also suggested membrane rigidity might play a crucial role in pore formation. However, no study has been conducted where the role of the actual plasma membrane (heterogeneous lipid environment)

in pore formation was elucidated. It is expected that shorter chain-length lipid bilayers or unstable lipid bilayers would be suitable targets for the toxins to rupture the membranes and punch holes. Furthermore, with the current development in cryo-EM, we were able to visualize the various intermediate conformations of small PFTs, like VCC, HlgAB, and α-HL in the lipidic environment (Chatterjee et al., 2025a; Chatterjee et al., 2025b; Mishra et al., 2023; Sengupta et al., 2021). Thus, we used the excellent opportunity to utilize cryo-EM to visualize α-HL in a real cellular environment, like RBC, where the composition of the cell membrane was highly heterogeneous, which affected the pore formation. On the other hand, the entire pore formation process is highly dynamic; therefore, making the plasma membrane slightly more rigid slowed down the barrel formation step of the toxin, allowing us to capture different conformational states of α-HL.

Cryo-EM has been a preferred choice to get different conformational states of a protein and to understand the transient cellular processes (Papasergi-Scott et al., 2024; Yoniles et al., 2024). Therefore, cryo-EM analysis of the toxins in the pre- and posthemolytic stages helped to obtain structures of different conformational states. This led us to conclude that the TM β-barrel should be stabilized to complete the cell lysis process.

A previous report suggested that only the H35R mutation resulted in larger oligomeric species than the common heptamer (Jursch et al., 1994). Our work demonstrated that slowing down the oligomerization process led to the generation of such intermediate and higher oligomeric species (octamer) from the wild-type protein. The octameric conformation existed in the prepore form, where the pore vestibule is blocked with a mixed density of lipid and unstructured stem domain. Previously, researchers have tried to understand the geometry of the half-oligomer or arc-like species of cholesterol-dependent cytolysins using negative-stain EM analysis or molecular dynamics simulations (Leung et al., 2014; Schaefer and Hummer, 2022), although these studies were limited to large PFTs. Hence, we are one of the first groups to report structural features of such arc-like oligomers of small β-PFTs in a cellular condition. The arc-like oligomer comprised a different number of promoters and appeared as an initial step of pore formation. Furthermore, the high-resolution structure of the octamers and arc-like species could be determined if we can increase the abundance of these species, either by modulating lipids, the sample preparation method, or generating mutants. Our mass spectrometry data also demonstrated a tendency for binding of α-HL toward the unstructured membrane part for pore formation and thereby inducing membrane lysis. Additionally, the lipid head group–binding site in the pore structure matches with crystal structure of pore conformations solved with MPD and detergent-exchanged $C_{14}$ PC (Liu et al.,

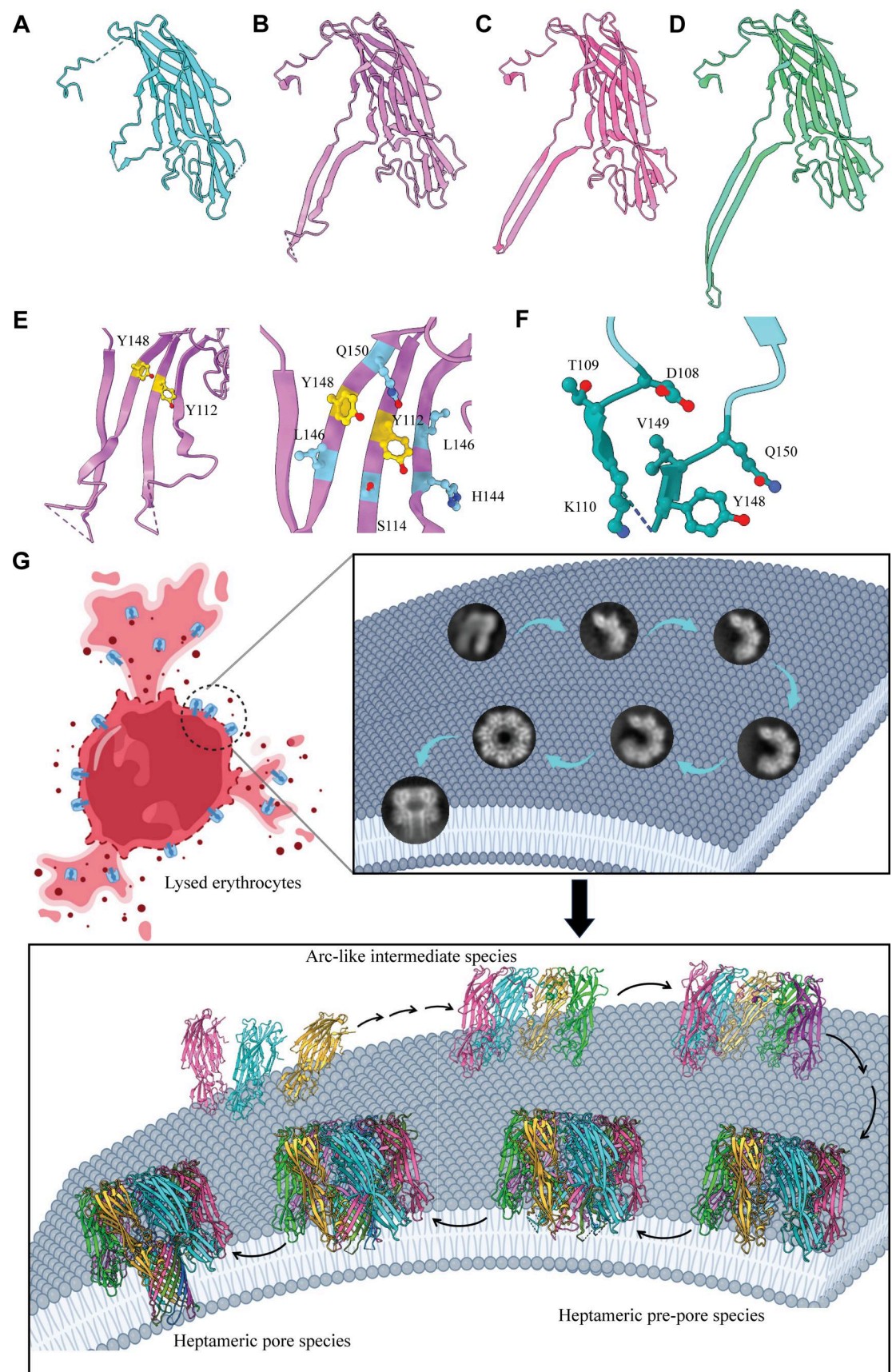

Figure 6. **Understanding the pore formation mechanism of α-HL in the cellular environment. (A–C)** Atomic models of single protomers of prepore states II–IV. **(D)** Atomic model of a single protomer of the pore structure. **(E)** Residues at the turning point of two β-strands of the prepore state III (pink color) are

highlighted in golden color, and the interacting partners (S114, H144, L146, and Q150) of Y148 and Y112 (in golden color) are shown in light blue color. **(F)** Residues in the non-TM portion of the stem domain are highlighted in dark green color. **(G)** Portion of lysed erythrocyte cells is zoomed to show 2D class averages of different oligomeric states (top) and their corresponding atomic model (bottom). (Hypothetical cell rupture and the membrane bilayer were created with https://BioRender.com).

2020; Tanaka et al., 2011). The RMSD comparison of pore structure in the cellular environment with liposome-derived pore structure showed a minimal change (Chatterjee et al., 2025b), whereas there was a change in the rim domain when compared to crystal structures, suggesting lipid-mediated changes in the rim domain (Song et al., 1996).

Furthermore, cryo-CLEM has become a popular tool to correlate the high-resolution data obtained using cryo-EM and fluorescence data (Fu et al., 2019; Pierson et al., 2024). Here, we used cryo-EM and fluorescence techniques separately on the same samples to characterize the morphological changes of cells that occurred during hemolysis. Additionally, we observed membrane protrusion induced by the toxin in HL-60 cells and RBCs. A similar blebbing and protrusion were also induced by other toxins, where enterotoxin had very negligible effect on reconstituted liposomes, whereas the toxin promoted blebbing from the membrane and cytolysis of MDCK cells in a receptor-dependent manner. The blebbing was reported as a phenomenon to resist cell lysis against streptolysin O attack on different cell lines (Babiychuk et al., 2011; Romero et al., 2017). However, as HL-60 cells and RBCs were susceptible to α-HL attack, this led to complete lysis of the cells. Therefore, our study indicated that cells attempted to protect the damage via plasma membrane protrusion, but the lysis occurred when the β-barrel was fully formed to perform the pore-mediated osmotic lysis.

In summary, we captured blebbing and protrusion of HL-60 cells and RBCs. Furthermore, we also captured the step-by-step oligomeric process of pore formation by α-HL during membrane rupture. Our study is one of the first studies where we visualized blebbing and protrusion of HL-60 cells and RBCs, as well as captured high-resolution structures of various intermediates of α-HL during pore formation (Fig. 6 G). This study is the only study to capture the entire pore formation process at high resolution. Furthermore, the involvement of toxins in the blebbed area can be identified in the future with the help of cryo-electron tomography. Also, we will be able to study how the plasma membrane's blebbing and protrusion help protect the cell from toxin attack.

## Materials and methods

### Cloning, overexpression, and purification of α-HL
The codon-optimized synthesized gene of α-HL (SG301D1; Biomatk) was converted to a wild-type gene using primers (Table S3: 179W and 200R) in a C-terminal 6× His-tagged pET26b (+) vector. The recloned wild-type α-HL gene was transformed and overexpressed into *E. coli* BL21 (DE3) strain cells (Novagen) using Luria Broth (HiMedia) growth media to an OD$_{600}$ of 0.6–0.8 at 37°C. For overexpression, isopropyl-β-D-thiogalactoside (HI-MEDIA) at 0.5 mM concentration was added to the cultured cells at 25°C for 12 h to overexpress the protein. The cultured cells were harvested at 3,825 × *g* for 20 min, and the pellet-containing

cells were further dissolved in lysis buffer (50 mM Tris-HCl buffer, pH 8, 150 mM NaCl, 1% glycerol), 7 mM imidazole, 2 mM phenylmethylsulfonyl fluoride, 1 mM DTT, and 0.8 mg/ml lysozyme. The dissolved pellet was incubated with lysis buffer at 4°C, followed by cell lysis using sonication for 30 min (20 kHz, amplitude 25%, Thermo Fisher Scientific). To remove cellular debris, the lysed cells were centrifuged, followed by centrifugation for 40 min at 18,328 × *g* in order.

For affinity-based protein purification, the Ni-NTA beads were equilibrated with the lysis buffer having 10 mM imidazole. The supernatant solution from the centrifugation was used for binding with equilibrated Ni-NTA beads for 3 h on a rotating platform at 4°C. The supernatant solution flowed through a gravity column. Then, the beads were washed with a wash buffer containing 50 mM Tris-HCl buffer, pH 8, 150 mM NaCl, 1% glycerol, and 20–60 mM imidazole (Sigma-Aldrich) in a gradient form. Finally, the His-tagged toxin was eluted using the 50 mM Tris-HCl buffer, pH 8, 150 mM NaCl, 1% glycerol, and 300 mM imidazole concentration.

For size-exclusion chromatography, 10/300 GL column (Cytiva) packed with Superdex 200 beads was preequilibrated with buffer compositions of 50 mM Tris-HCl buffer, pH 8, 150 mM NaCl, and 1% glycerol. The Ni-NTA–purified toxin was loaded into the 10/300 Gl column. The eluted sample from the peak fraction was concentrated using an Amicon Ultra centrifugal filter with a molecular cutoff of 10 kDa (Amicon Ultra-4 Centrifugal filter unit 10 K, Merck). The concentrated sample was further loaded into 10% SDS-PAGE (Coomassie Brilliant Blue R 250, Merck) to confirm its purity. Precision Plus Protein Dual Color Standards (Bio-Rad) were used as a protein molecular weight marker in SDS-PAGE.

### Site-directed mutagenesis and truncated clone generation from wild-type α-HL
The double primer method of site-directed mutagenesis was performed to get a point mutant (A200R) from the synthesized double-mutant (W179A-R200A) α-HL gene. The polymerase chain reaction (PCR) was conducted for 30 cycles after the initial denaturation step at 98°C for 2 min, with a follow-up thermal denaturation at 98°C for 8 s, primer annealing at 58–65°C for 30 s, DNA synthesis at 72°C for 3.5 min, and a final extension at 72°C for 5 min. The PCR products were subjected to digestion using DpnI, followed by inactivation at 80°C for 20 min. The annealed PCR products were transformed into *E. coli* (DH5α) ultracompetent cells. Plasmids were isolated from single colonies using QIAprep Spin Miniprep Kit (QIAGEN) and sequenced for the identification of positive mutants. For the N-terminal deleted truncation (A1-T9 amino acids) construct generation from wild type, PCR was performed with an initial thermal denaturation at 98°C for 30 s, a follow-up primer annealing at 58–65°C for 30 s, DNA synthesis at 72°C for 5 min, and a final extension at 72°C for

10 min. The PCR products were ligated using the ligation mixture. The ligated DNA mixture was purified and transformed into *E. coli* DH5α cells. The generated N-terminal truncated clone was further confirmed by DNA sequencing. The N-terminal truncated clone and mutants of α-HL were overexpressed and purified using the same method described for wild-type α-HL.

### Cell death analysis using flow cytometry

HL-60 cells (ATCC, https://www.atcc.org/products/ccl-240) were maintained in Iscove's modified Eagle's medium (Sigma-Aldrich), supplemented with 20% fetal bovine serum and 1% antibiotic in a 37°C and 5% $CO_2$ incubator. On the day of toxin treatment, ~$1*10^6$ HL-60 cells per well were taken and seeded in a 12-well plate. The cells were then treated with 100 nM of α-HL toxin concentration and incubated for one and a half hours in a 37°C and 5% $CO_2$ incubator. Simultaneously, untreated healthy HL-60 cells were also incubated for the same duration as a control. Following treatment with α-HL toxin, the cells were harvested by centrifugation and washed with the wash buffer containing 1× phosphate-buffered saline (PBS) and 3% BSA. After washing, the cells were incubated with PI (Cat#40017; Biotium) for ~30 min at 37°C under constant shaking. After labeling, the cells were washed again and analyzed by flow cytometry using the BD Aria III instrument. For analyzing the cells by flow cytometry, 585/42-nm band-pass filter was used for PI.

### Confocal imaging

To analyze the cytotoxic impacts, cultured HL-60 cells were incubated with toxin at a 100 nM concentration of α-HL and were visualized under Zeiss ELYRA 7 ApoTome imaging using a 40× oil immersion objective (Plan-Apochromat 40×/1.4 Oil DIC, numerical aperture [NA]: 1.40) with Lattice SIM super-resolution microscope in the presence of lipophilic membrane dye Nile red (Cat#72485; Sigma-Aldrich) and DNA staining dye DAPI (Cat#422801; BioLegend). An excitation laser source at 405 and 540 nm was used for DAPI and Nile red fluorescence imaging, respectively. The emission band-pass filter for DAPI and Nile red was 480–510 and 565–610 nm, respectively. The 3DSIM settings using ZEN core software (ZEISS India) corresponded to a periodic grid of period 23 for 488 nm and 28 for 555 nm, 3 rotations, and 5 translations. Leica Falcon SP8 (40× oil immersion objective, Leica HC PL APO 40×/1.30 Oil CS2, numerical aperture (NA): 1.40, LAS X software) was also used to perform all the confocal data for fluorescence imaging of RBC. Nile red was added to label the membrane just before imaging. Nile red provided a solvatochromic diverse emission spectrum after binding to the rigid part of the RBC plasma membrane (Kucherak et al., 2010). For checking the change in morphology, the excitation was set to 540 nm, and the emission was set at a range of 565–610 nm, whereas to check rigidity, the excitation was set to 540 nm, and emission was set at a range of 525–565 nm, and another range of 585–610 nm. Data were visualized and analyzed using Fiji/ImageJ (Schindelin et al., 2012).

### Hemolysis assay

About 1 ml of whole blood was isolated from *Oryctolagus cuniculus* and was centrifuged multiple times (6–8) at 201 × *g* for 10 min

using 1× sodium PBS containing 137 mM sodium chloride (Qualigens), 1.8 mM potassium dihydrogen orthophosphate (Qualigens), 10 mM disodium hydrogen orthophosphate (SD Fine Chemicals), and 2.7 mM potassium chloride (Sigma-Aldrich) to isolate plasma component-free intact RBCs for the hemolysis assay with α-HL. 1× PBS buffer solution was used to prepare a 10% RBC solution. As a positive control for hemolysis, 1% Triton X-100 detergent solution was used, whereas for a negative control, 1× PBS buffer was used. RBCs were treated with different concentrations (μM) of α-HL and incubated in constant shaking conditions (0.057 × *g*) at 37°C and 10°C for a duration of 15 min. The optical density (OD) of the RBCs after toxin treatments under different conditions was measured at 620 nm using a Varioskan Flash microplate reader (Thermo Fisher Scientific).

### Circular dichroism spectroscopy

The melting studies of the toxin were performed with large unilamellar vesicles, which were made up of 1-palmitoyl-2-oleoyl-sn-glycero-3-phosphatidylcholine (POPC). POPCs stored at different temperatures (37°C and 4°C) were selected for performing circular dichroism spectroscopy using a JASCO J-715 spectrophotometer. The ellipticity was monitored from 35°C to 90°C with the wavelength scans from 195 to 260 nm.

### Mass spectrometry analysis

To isolate the lipids, the pellet and supernatant fractions of the RBC-α-HL complex were treated with chloroform, followed by drying of the samples. The dried samples were further dissolved in chloroform and analyzed using a MALDI time-of-flight mass spectrometer.

### Negative staining sample preparation and visualization by TEM

Three-microliter sample of monomeric toxin was incubated with washed rabbit erythrocytes. The samples after centrifugation, and only erythrocytes, were applied to a thin film of carbon-coated copper grids (Carbon Film 300 Mesh, Copper, Electron Microscopy Sciences) at room temperature, which was glow-discharged for 30 s, 20 mA freshly. These samples were allowed to settle down on the grid for 1 min, before the removal of excess buffer. Immediately after that, 1% uranyl acetate (Polysciences, Inc.) was used for staining the grid surface containing the sample, followed by blotting to remove excess stain from the grid. These negatively stained samples were then visualized under a 120 kV Talos L120C transmission electron microscope equipped with a bottom-mounted Ceta camera (4,000 × 4,000 pixels) at magnification ranging between ×2,300 and ×73,000. 20 images of the whole cell were collected for both the control and toxin-treated conditions at low magnification. Some of the images were further taken at high magnification to further characterize like morphology and membrane changes of RBCs.

### Data processing for NS-TEM micrographs

Different particle projections were manually picked from 2D raw NS-TEM micrographs using e2projectmanager.py (EMAN2.91 [Tang et al., 2007]), followed by the extraction of particles with a 120-pixel box size and a processing 56 Å mask diameter from

raw micrographs using e2boxer.py. An average of 6,000–7,000 particles from each dataset were further classified into 2D-class averages using reference-free 2D classifications in EMAN2.91 (Tang et al., 2007) program.

## Cryo-EM sample preparation

R1.2/1.3 300 mesh copper grids (Quantifoil; Electron Microscopy Sciences) were glow-discharged for 90 s at 20 mA before the cryo-sample freezing process. HL-60 cells were washed using PBS to remove FBS from the solution, and then incubated with α-HL (1 mg/ml) at room temperature for 30 min. For the post-hemolytic RBC-α-HL complex, α-HL (1 mg/ml) was mixed with rabbit erythrocytes and incubated at 37°C for 15 min. After incubation, we centrifuged the mixture at 18,328 × g for 1 min. Three microliters of supernatant solution from that cell–protein mixture was applied onto the freshly glow-discharged grids and immediately blotted for 8 s with a blot force of positive 25 using an FEI Vitrobot Mark IV plunger. The grid was plunged into the liquid ethane instantly after blotting. For the prehemolytic RBC-α-HL complex, we instantly centrifuged the mixture at 18,328 × g for 1 min at 4°C. and then underwent the same procedure as posthemolysis sample preparation.

## Cryo-EM data collection

Cryo-EM data were acquired using a 200 kV Talos Arctica transmission electron microscope (Thermo Fisher Scientific) equipped with a K2 Summit Direct Electron Detector (Gatan, Inc.). Movies were recorded automatically using Latitude-S (Kumar et al., 2021) (Digital Micrograph—GMS 3.5) with a total dose of 50 e$^-$/Å$^2$, with an exposure time of 8 s distributed for 20 frames at a magnification of 54,000× at the effective pixel size of 0.92 Å. A total of 877 micrographs were acquired for the HL-60 cell–α-HL protein mixture condition. A total of 1,635 and 2,691 micrographs were acquired for the posthemolytic and prehemolytic RBC-α-HL protein complexes, respectively. The whole-cell imaging was done at 11,000×–22,000× magnification to visualize the cell morphology. For correlative imaging, almost 50 whole-cell images were collected at a very low magnification (84×), where many cells were visible. Then, the best-matched cell with our confocal data was collected at high magnification.

## Cryo-EM data processing

Single-particle analysis was performed using the cryoSPARC 4.5 (Punjani et al., 2017). The drift and gain corrections of the collected movies were performed with Patch Motion Correction, followed by an estimation of Contrast Transfer Function (CTF) parameters using Patch CTF estimation. Micrographs were subjected to a curating exposure with a resolution cutoff of 8 Å, where we removed micrographs of resolution worse than 8 Å. The density map generated from the crystal structure (PDB ID: 7AHL) (Song et al., 1996) was used to create a template, and the different views were provided as templates for template-picker–based picking. During the inspection, we have kept the best higher NCC score and lower histogram values for more accurate picking of desired particles. The particles were extracted with a box size of 280 pixels (pixel size of 0.92 Å) for both samples. Three rounds of reference-free 2D classification were run each

time to remove wrongly picked or wrongly aligned particles. The best 2D classes were selected for 3D classification with C7 (for heptameric species) or C1 (for octameric and arc-like species) symmetry. The ab initio model generated from 2D class averages and the density map from the crystal structure (PDB ID: 7AHL) (Song et al., 1996) were filtered to 40 Å to use as a reference for heterogeneous refinement and nonuniform refinement. We performed heterogeneous refinement for heptameric conformations with C7 symmetry to further classify the particles. The best classes were subjected to nonuniform refinement using C7 symmetry. High-resolution heptameric structures were also processed using C1 symmetry. For octameric and pentameric arc-like species, the selected best particles were selected from 2D class averages, for ab initio reconstruction (without imposing any symmetry), followed by nonuniform refinement (C8 symmetry for octamer and C1 symmetry for pentamer). The high-resolution (<4 Å) heptameric conformations were subjected to reference-based motion correction, followed by nonuniform refinement and local refinement, which includes the estimation of spherical aberration, anisotropic magnification, per-particle defocus, and CTF refinement. Global resolution of Fourier shell correlation was estimated at the threshold of 0.143 (Scheres and Chen, 2012), and the estimation of a local resolution was performed with the local resolution estimation tool in cryoSPARC.

## Model building and structure refinement

PDB ID: 7AHL (Song et al., 1996) was fitted to the cryo-EM maps in UCSF ChimeraX to use as a template for atomic model generation for heptameric pore and prepore conformations. Model building was done using Coot (Emsley and Cowtan, 2004) and Phenix (Liebschner et al., 2019). The phenix.real_space_refine models and part of the PC head group were fitted to the cryo-EM map of pore conformation using UCSF ChimeraX (Goddard et al., 2018).

## Analysis and visualization

The cryo-EM maps and atomic models were validated using Phenix, and EMRinger (Barad et al., 2015) and MolProbity (Williams et al., 2018), respectively. The structural comparisons between different conformations were performed using the UCSF ChimeraX "MatchMaker" tool. The detailed structural analysis was performed using ChimeraX (Goddard et al., 2018) and PyMOL.

## Data deposition

Electron density maps and their corresponding coordinate files were deposited to the Electron Microscopy Data Bank (EMDB) and PDB, respectively. Data identifiers at EMDB and PDB of the α-HL heptameric pore isolated from the posthemolytic state are EMD-63626 and 9M4P, respectively. Data identifier at PDB of the PC head group–bound α-HL heptameric pore structure in the presence of RBC is 9UFA. Data identifiers at EMDB and PDB of the α-HL heptameric prepore state IV isolated from the prehemolytic state are EMD-63620 and 9M4A, respectively. Data identifiers at EMDB and PDB of the α-HL heptameric prepore state III isolated from the prehemolytic state are EMD-63781 and 9MBX, respectively. Data identifiers at EMDB and PDB of the

α-HL heptameric prepore state II isolated from the prehemolytic state are EMD-65040 and 9VFW.

## Online supplemental material

Fig. S1 shows characterization of purified recombinant toxin and cryo-EM analysis with HL-60 cells and α-HL–mediated effect on RBCs. Fig. S2 demonstrates NS-TEM and cryo-EM analysis of prehemolytic and posthemolytic states of RBCs. Fig. S3 provides cryo-EM sample preparation for α-HL–treated RBCs and cryo-EM analysis of oligomeric states of α-HL obtained from the posthemolytic RBC–toxin complex. Fig. S4 shows cryo-EM data processing pipeline of α-HL heptamers obtained from the prehemolytic RBCs-α-HL complex. Fig. S5 shows comparison of barrel and lipid densities of different prepore states. Fig. S6 provides cryo-EM analysis of α-HL arc-like and octameric species in prehemolytic RBCs. Table S1 shows cryo-EM data collection, image processing, and refinement for pre- and posthemolysis α-HL conformations. Table S2 shows cryo-EM map and model validation for membrane and α-HL complex. Table S3 shows primer information. Video 1 shows control HL-60 cell. Video 2 shows protrusion from HL-60 cell membrane and damage to the cell after toxin treatment. Video 3 shows changes in RBC membrane during toxin-mediated hemolysis. Videos 4 and 5 show RBC fragmentation after toxin treatment. Video 6 shows protrusion from RBC membrane after toxin treatment. Data S1 shows values corresponding to the bar graph related to Fig. 1 C. Data S2 shows values corresponding to the plot related to Fig. 1 D. Data S3 shows values corresponding to the bar graph related to Fig. 2 D. Data S4 shows values corresponding to the bar graph related to Fig. 2 G. Data S5 shows values corresponding to the plot related to Fig. 4 A. Data S6 shows values corresponding to the plot related to Fig. 4 C. Data S7 shows values corresponding to the bar graph related to Fig. 5 G. Data S8 shows values corresponding to the bar graph related to Fig. 5 H. Data S9 shows values corresponding to the bar graph related to Fig. 5 I.

## Data availability
The data are available upon reasonable request.

# Ethics statement
## Ethics approval
Rabbit plasma was obtained after approval of the Institutional Animal Ethics Committee, Indian Institute of Science, Bangalore (CAF/Ethics/881/2022).

# Acknowledgments
We acknowledge Department of Biotechnology (DBT), Department of Science and Technology (DST), Council of Scientific and Industrial Research (CSIR), and Ministry of Education, India, for funding and cryo-EM facility at IISc, Bengaluru.

We acknowledge DBT-BUILDER Program (BT/INF/22/SP22844/2017) and DST-FIST (SR/FST/LSII-039/2015) for National Cryo-EM Facility at IISc, Bengaluru. S. Dutta thanks DBT (File No. BT/PR54674/BMS/85/475/2024) for financial support. S. Dutta acknowledges the high-performance computing cluster "Beagle" at the Biological Sciences Division, IISc. A. Chatterjee acknowledges CSIR (Grant No. 09/0079(13652)/2022-EMR-I) for doctoral fellowship. S. Dutta acknowledges Science and Engineering Research Board, India (Grant No. CRG/2022/002674), for financial support. S. Dutta thanks STR/2022/000006 SERB-STAR award for financial support. We thank the mass spectrometry and bioimaging facilities at IISc. Open Access funding provided by the Indian Institute of Science.

Author contributions: Arnab Chatterjee: conceptualization, data curation, formal analysis, investigation, methodology, validation, visualization, and writing—original draft, review, and editing. Anupam Roy: formal analysis, investigation, methodology, validation, visualization, and writing—review and editing. Partho Pratim Das: data curation, formal analysis, methodology, resources, software, visualization, and writing—original draft, review, and editing. Debajyoti Chakraborty: data curation, formal analysis, methodology, software, validation, visualization, and writing—review and editing. Bartika Ghoshal: resources. Siddharth Jhunjhunwala: methodology, resources, and writing—review and editing. Somnath Dutta: conceptualization, data curation, funding acquisition, investigation, project administration, resources, supervision, validation, visualization, and writing—original draft, review, and editing.

Disclosures: The authors declare no competing interests exist.

Submitted: 22 June 2025

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

# Supplemental material

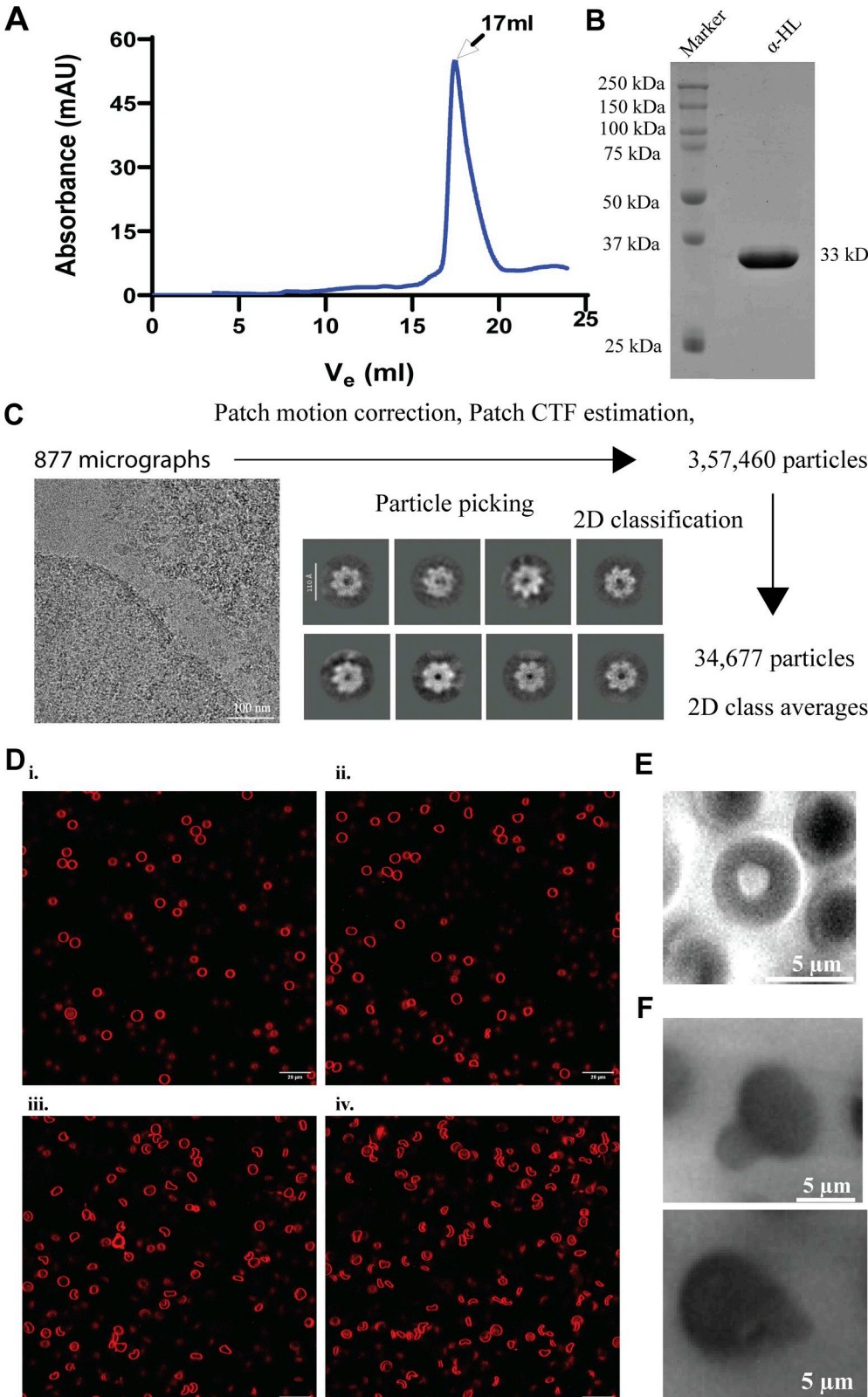

Figure S1. **Characterization of purified recombinant toxin and cryo-EM analysis with HL-60 cells and α-HL–mediated effect on RBCs. (A)** Size-exclusion chromatography profile of purified α-HL monomer. **(B)** SDS-PAGE of SEC-purified recombinant α-HL monomer. **(C)** Data processing pipeline of α-HL in the presence of HL-60 cells. **(D)** Toxin-incubated RBC sample was imaged under a confocal microscope at different time points: (i) just after toxin addition; (ii) starting point of membrane damage after 2 min; (iii) lysis initiation state after 5 min; and (iv) lysed cells after 10 min. The cell membranes (in red color) were stained using Nile red dye ($\lambda_{excitation}$ = 540 nm, $\lambda_{emission}$ = 565–610 nm). **(E)** Control RBC. **(F)** Bright-field imaging of toxin-treated RBC showing protrusion from the membrane. Source data are available for this figure: SourceData FS1.

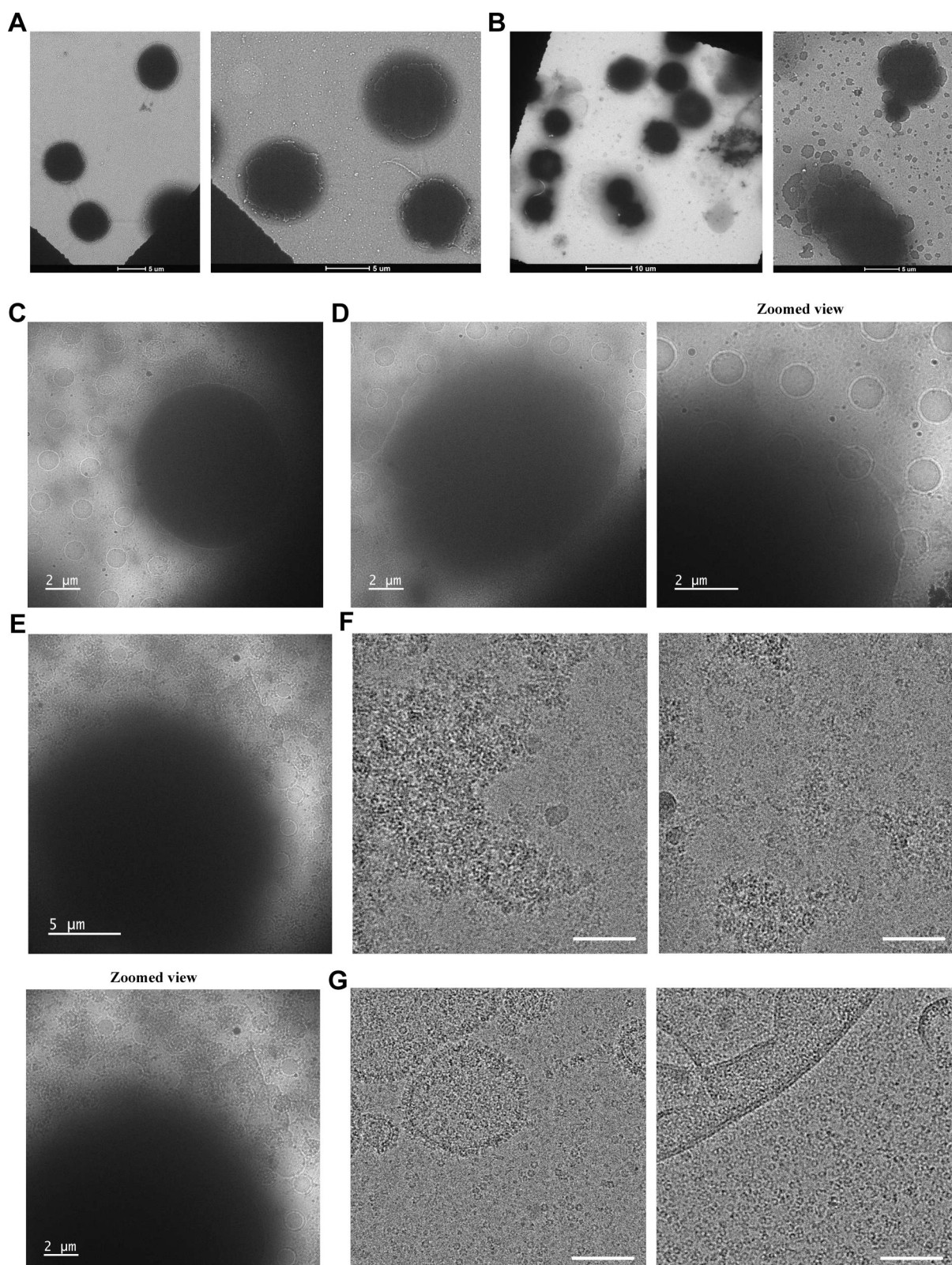

Figure S2.    **NS-TEM and cryo-EM analysis of prehemolytic and posthemolytic states of RBCs. (A)** NS-TEM images of control RBCs (low-magnification image at the left side and high-magnification image at the right side). **(B)** NS-TEM images of toxin–RBCs (low-magnification image at the left side and high-magnification image at the right side). 30 images were collected from different regions in triplicate to validate this phenomenon. **(C)** Cryo-EM image of control RBC (entire cell). **(D)** Cryo-EM image of RBC (entire cell) in prehemolytic condition and its zoomed image. **(E)** Cryo-EM image of RBC (entire cell) in post-hemolytic condition and its zoomed view. **(F and G)** Cryo-EM micrographs of toxin-containing membranes, either in lysed form or in vesicular form, outside the RBCs in post- and prehemolytic conditions, respectively (scale bar: 100 nm).

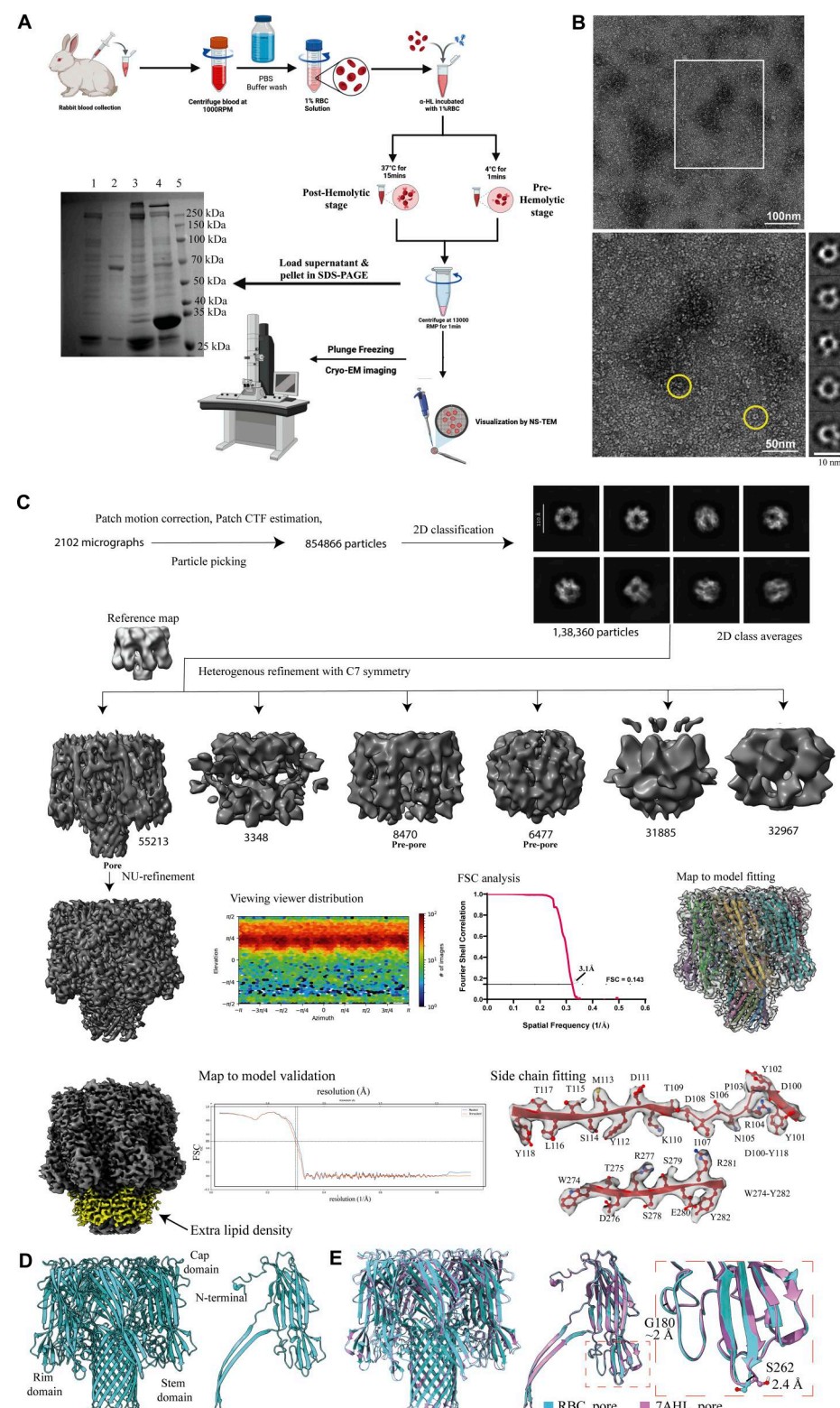

Figure S3. **Cryo-EM sample preparation for α-HL–treated RBCs and cryo-EM analysis of oligomeric states of α-HL obtained from posthemolytic RBC–toxin complex. (A)** Sample preparation procedure of α-HL–treated RBCs (created with BioRender.com). SDS-PAGE gel showing lanes corresponding to untreated RBC pellet—lane 1; untreated RBC supernatant—lane 2; toxin-treated RBC pellet—lane 3; toxin-treated RBC supernatant—lane 4; and protein marker—lane 5. **(B)** NS-TEM analysis of the supernatant fraction of α-HL–treated lysed RBCs. An enlarged view of the white squared region is shown, the oligomers are encircled in yellow color, and the 2D class averages of those particles are shown. **(C)** Cryo-EM data processing pipeline of α-HL oligomers present in the posthemolytic condition. The extra membrane density is shown in yellow, and the map in gray. **(D)** Atomic model of heptameric pore structure (in cyan). **(E)** Comparison of pore structure solved in the presence of RBCs (cyan) with the pore structure solved in the presence of detergent (pink). The shifts in S262 and G180 residues in the rim domain are 2.4 Å and ~2 Å, respectively.

Cryo-EM data processing pipeline of heptameric α-HL in pre-hemolytic condition

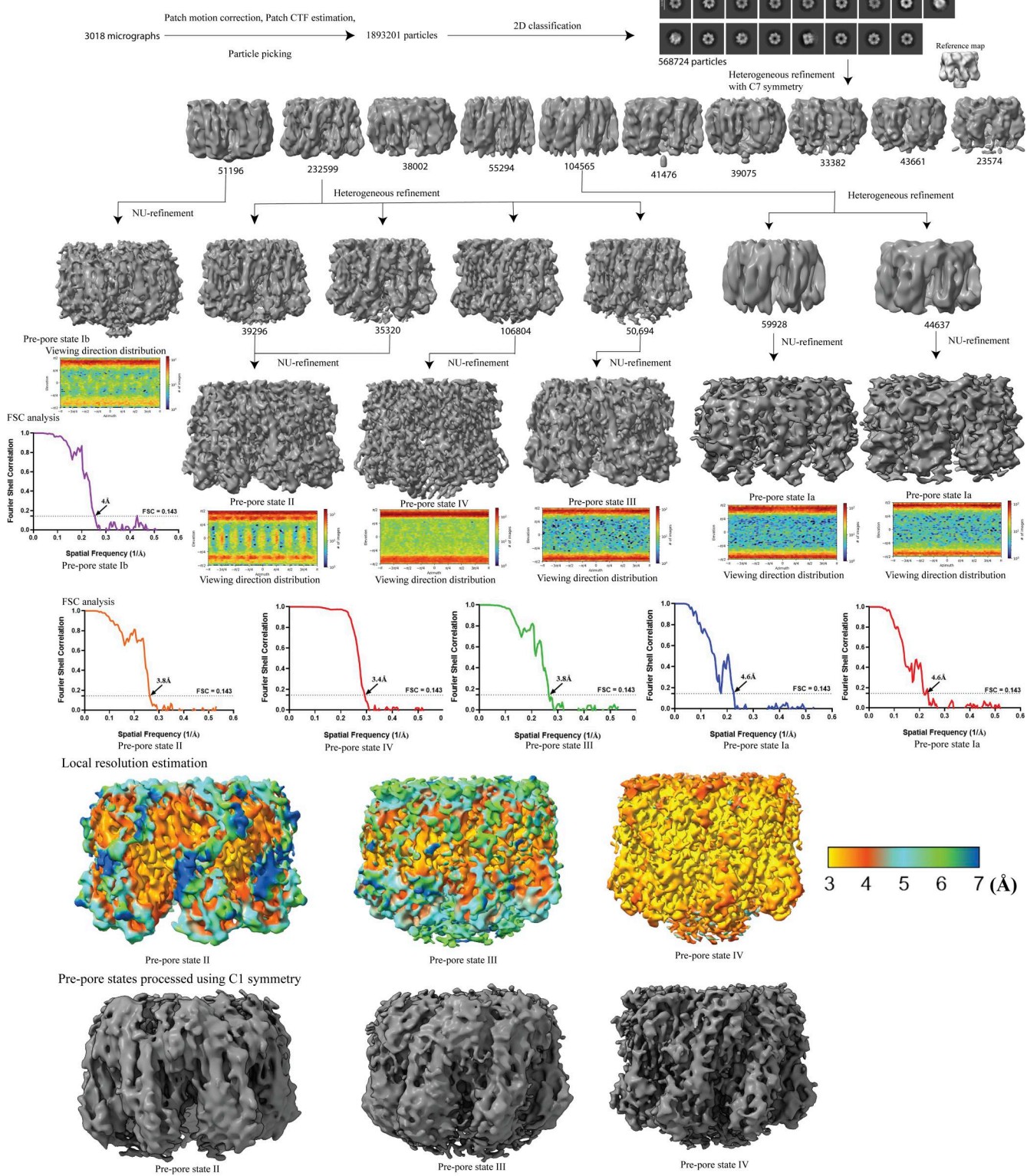

Figure S4. **Cryo-EM data processing pipeline of α-HL heptamers obtained from prehemolytic RBC–α-HL complex.** Source data are available for this figure: SourceData FS4.

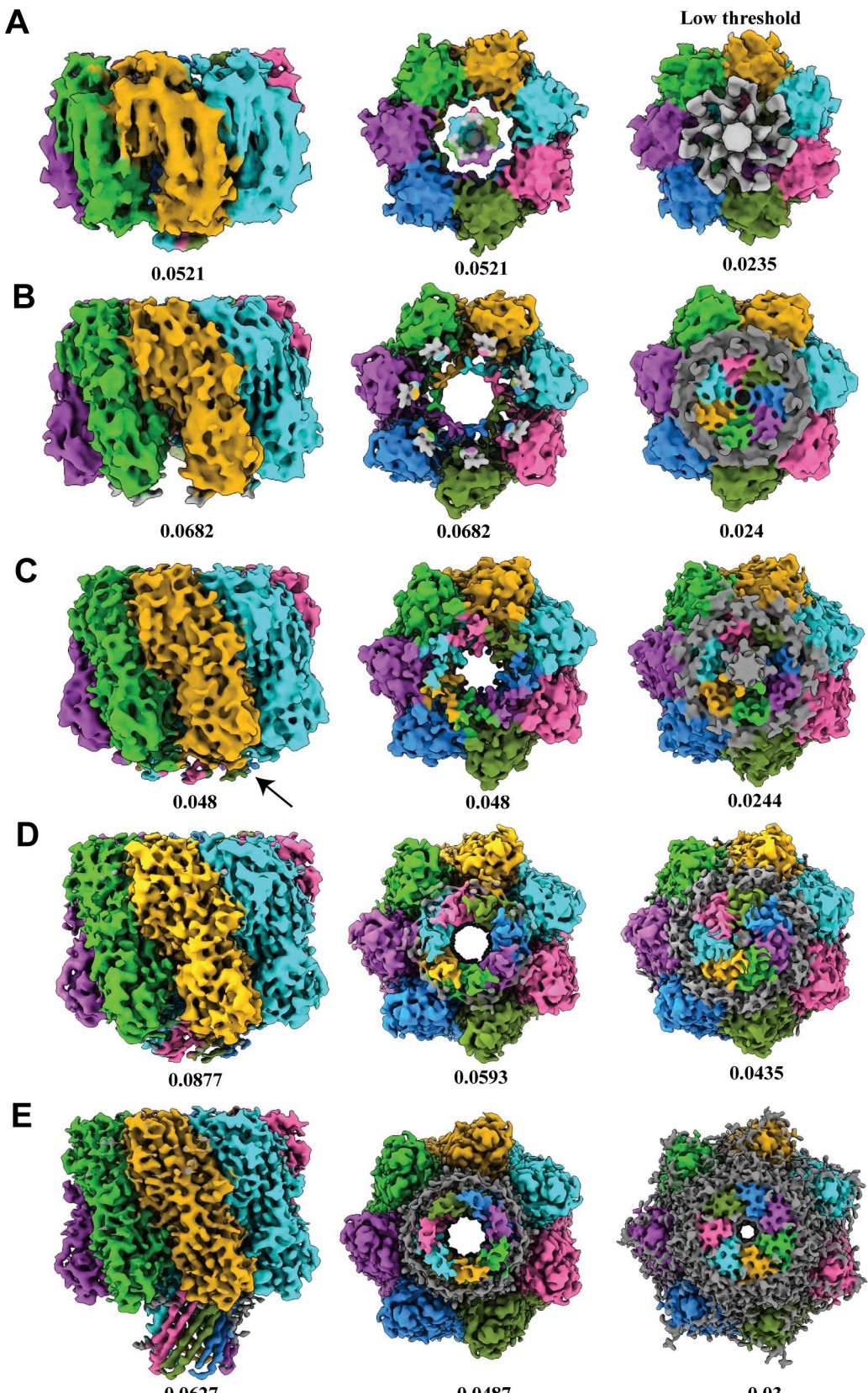

Figure S5.  **Comparison of barrel and lipid densities of different prepore states. (A–E)** Prepore states IV-Ia and pore state at different thresholds (A—prepore Ib; B—prepore II; C—prepore III; D—prepore IV; E—pore). Seven protomers are colored in different colors, and the lipid densities near the stem region at a lower threshold are shown in gray. The threshold values are mentioned at the bottom of the maps. The extra TM region of the prepore state III is shown using a black arrow.

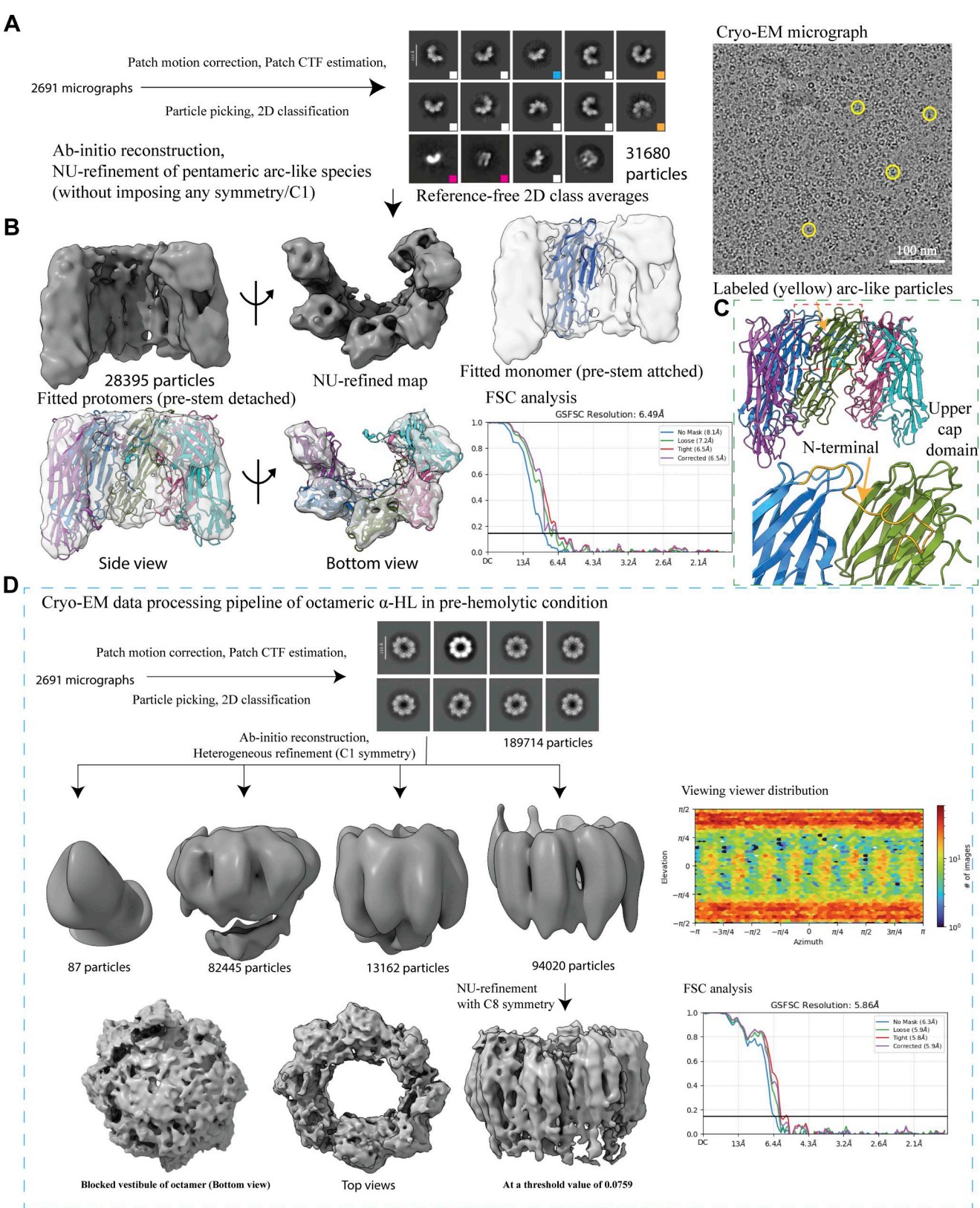

Figure S6.   **Cryo-EM analysis of α-HL arc-like and octameric species in prehemolytic RBCs. (A)** Cryo-EM data processing pipeline of α-HL arc-like species. Different sizes of arcs are highlighted in different square boxes (pink—dimer; blue—4-mer; white—5-mer; and orange—6-mer). **(B)** 3D refinement of the pentameric/5-mer species, and model fitting different protomers shown in different colors in the EM map (gray color). The prestem detached protomer was used from our pore structure, and for the prestem attached monomer (blue color), the PDB ID: 4YHD structure was used. **(C)** Orientation of the pentameric structure. The red-box region shows the zoomed view of the N-terminal region. In the zoomed view, the N terminus of one of the protomers is highlighted in golden rod color, showing its interaction with the upper cap domain. **(D)** Cryo-EM data processing pipeline of α-HL octamers in prehemolytic RBCs.

Video 1.  **Control HL-60 cell.**

Video 2.  **Protrusion from HL-60 cell membrane and damage to the cell after toxin treatment.**

Video 3.  **Changes in RBC membrane during toxin-mediated hemolysis.**

Video 4.  **RBC fragmentation after toxin treatment.**

Video 5.  **RBC fragmentation after toxin treatment.**

Video 6.  **Protrusion from RBC membrane after toxin treatment.**

**Provided online are Table S1, Table S2, and Table S3. Table S1 shows cryo-EM data collection, image processing, and refinement for pre- and posthemolysis α-HL conformations. Table S2 shows cryo-EM map and model validation for membrane and α-HL complex. Table S3 shows primer information. Data S1 shows values corresponding to the bar graph related to Fig. 1 C. Data S2 shows values corresponding to the plot related to Fig. 1 D. Data S3 shows values corresponding to the bar graph related to Fig. 2 D. Data S4 shows values corresponding to the bar graph related to Fig. 2 G. Data S5 shows values corresponding to the plot related to Fig. 4 A. Data S6 shows values corresponding to the plot related to Fig. 4 C. Data S7 shows values corresponding to the bar graph related to Fig. 5 G. Data S8 shows values corresponding to the bar graph related to Fig. 5 H. Data S9 shows values corresponding to the bar graph related to Fig. 5 I.**

