## [Peer Review File · The Journal of Cell Biology]

Stepwise assembly of α -hemolysin from intermediates to the mature pore in native erythrocytes

Arnab Chatterjee, Anupam Roy, Partho Das, Debajyoti Chakraborty, Bartika Ghoshal, Siddharth Jhunjhunwala, and Somnath Dutta

Corresponding Author(s): Somnath Dutta, Indian Institute of Science Bangalore

Review Timeline:

Submission Date:	2025-06-22
Editorial Decision:	2025-08-20
Revision Received:	2025-09-22
Editorial Decision:	2025-11-17
Revision Received:	2025-11-22

Monitoring Editor: Karin Reinisch

Scientific Editor: Dan Simon

Transaction Report:

DOI: <https://doi.org/10.1083/jcb.202506129>

August 20, 2025

Re: JCB manuscript #202506129

Somnath Dutta
Indian Institute of Science Bangalore

Dear Prof. Dutta,

Thank you for submitting your manuscript entitled "Cryo-EM Captures the Stepwise Assembly of α -Hemolysin Leading to Erythrocyte Membrane Lysis." The manuscript was assessed by expert reviewers, whose comments are appended to this letter. We invite you to submit a revision if you can address the reviewers' key concerns, as outlined here.

You will see that you overall the reviewers are enthusiastic about your study and feel it is suitable for JCB. They've provided constructive comments asking for a few experiments, additional analysis of the current data, clarifications and more details, and changes to figures to improve presentation. All of these should be addressed in some way in a revised manuscript. A particularly important issue is the need for a thorough comparison with the structure of α -Hemolysin on liposomes reported in your recent paper and explanation of the differences seen in this study with erythrocyte membranes.

GENERAL GUIDELINES:

Text limits: Character count for an Article is < 40,000, not including spaces. Count includes title page, abstract, introduction, results, discussion, and acknowledgments. Count does not include materials and methods, figure legends, references, tables, or supplemental legends.

Figures: Articles may have up to 10 main text figures. Figures must be prepared according to the policies outlined in our Instructions to Authors, under Data Presentation, <https://jcb.rupress.org/site/misc/ifora.xhtml>. All figures in accepted manuscripts will be screened prior to publication.

Supplemental information: There are strict limits on the allowable amount of supplemental data. Articles may have up to 5 supplemental figures. Up to 10 supplemental videos or flash animations are allowed. A summary of all supplemental material should appear at the end of the Materials and methods section.

Please note that JCB now requires authors to submit Source Data used to generate figures containing gels and Western blots with all revised manuscripts. This Source Data consists of fully uncropped and unprocessed images for each gel/blot displayed in the main and supplemental figures. For assays performed using capillary electrophoresis and/or immunoassay-based detection, authors should instead provide the electropherogram graph(s) for each experiment, plotting fluorescence/chemiluminescence intensity vs. molecular weight/size. Please be sure to provide one Source Data file for each figure gels, blots, and/or capillary electrophoresis assays along with your revised manuscript files. File names for Source Data figures should be alphanumeric without any spaces or special characters (i.e., SourceDataF#, where F# refers to the associated main figure number or SourceDataFS# for those associated with Supplementary figures). For traditional gels and blots, the lanes of the gels/blots should be labeled as they are in the associated figure, the place where cropping was applied should be marked (with a box), and molecular weight/size standards should be labeled wherever possible. For capillary electrophoresis assays, each trace in the graph should be color-coded and labeled to indicate which protein, gene, or sample is being measured (please try to avoid red/green combinations to accommodate our color-blind readers).

The typical timeframe for revisions is three to four months. If you anticipate any difficulties in meeting this aforementioned

revision time limit, please contact us and we can work with you to find an appropriate time frame for resubmission. Please note that papers are generally considered through only one revision cycle, so any revised manuscript will likely be either accepted or rejected.

Thank you for this interesting contribution to Journal of Cell Biology. You can contact us at the journal office with any questions at cellbio@rockefeller.edu.

Sincerely,

Karin Reinisch, PhD
Monitoring Editor
Journal of Cell Biology

Dan Simon, PhD
Scientific Editor
Journal of Cell Biology

Reviewer #1 (Comments to the Authors (Required)):

Chatterjee et al. conducted a comprehensive functional and structural analysis of α -Hemolysin on erythrocyte membranes. They demonstrated that hemolysin oligomerizes upon incubation with HL-60 cells and erythrocytes. High-resolution structures of the mature pore form were obtained, revealing potential visualization of the phosphatidylcholine (PC) headgroup, along with a minor conformation representing the pre-pore state. To capture additional intermediate pre-pore stages, the authors lowered the incubation temperature, which allowed them to observe several transitional pre-pore conformations. These intermediates exhibit significant differences in the β -barrel region, suggesting a sequential transition among states. Additionally, pre-pore assemblies with 4-mer, 5-mer, and up to 8-mer configurations were identified, displaying varying degrees of openness (arc pre-pores), illustrating the progression from monomer binding to heptameric pre-pore assembly.

Overall, the work is well performed and described; the structural analysis is sound. The figures presentation should be improved, though. I have several comments:

1. Why the pore-form and pre-pore form hemolysin oligomer are mostly detached from membrane in the cryo-EM sample?
2. Figure S1-C: 'The cryo-EM micrographs depicted the presence of circular toxins on the cell membrane.' Could the authors clarify whether the circular toxin structures observed are present on the vesicles or free in solution in the samples containing HL-60 cells?
3. The 2D class averages are not convincing. Given the diversity of proteinous densities in the raw micrograph, it is unclear whether the 2D average is actually from the designated toxins. Please provide better 2D class averages and indicate the possible pore particles. Otherwise, the authors may consider to remove this 2D class averages.
4. The HL-60 derived vesicles do not have much toxins. Why? One suggestion would be to re-add an excessive amount of toxins to the isolated HL-60-derived vesicles and then check whether hemolysin decorates these vesicles.
5. Figure 2B, please indicate the possible rings in the B(iii) and B(v). A higher magnification or zoom-in view will be helpful.
6. Figure 2E, the "correlative microscopy" data is not convincing, unless the authors statistically quantify the distortions and check whether the statistics in cryo-EM matches with fluorescent microscopy. The intact RBC and toxin-treated RBC are both too thick to be well-imaged by cryo-EM; the morphology changes under the cryo-EM conditions could be due to other physical changes during plunge freezing as well. The fluorescent imaging data is convincing and well presented to indicate the morphology change.
7. L254- prepore state along the membrane? Or also in the solution? In Figure 4D, there are particles in both solution state and membrane bound state. Are the membrane-bound structure in a compositional and conformational different state from solution oligomers?
8. L266, the presence of both unsaturated and saturated lipids with different chain lengths, leads to instable membrane bilayer? Any reference for this?
9. Figure S5 resolution distribution plot should be better presented. The color is confusing.
10. In Figure S5, the pre-pore form without C7 symmetry should be performed as well. Any twist in structures?
11. Have the authors attempted incubating HL-60 cells with α -hemolysin at 4{degree sign}C, similar to the conditions used with RBCs? Such an approach could help capture the pre-pore state or yield supernatant with less intracellular material, thereby reducing background noise in micrographs.

12. α -Hemolysin (α -HL) is known to form pores not only in leukocytes and erythrocytes but also in other cell types such as epithelial and endothelial cells. Have the authors investigated the pore-forming activity of α -HL in these additional cell types?
13. Figure 3 C: It would be beneficial to include a label indicating the size of the pore.
14. Figure S4C: 'a clear extra density was visible surrounding the TM segment and lower rim domain. We speculated that density was coming from the membrane components and was found to be nicely decorating the membrane binding sites of the toxin.' It would be helpful to highlight the regions of additional density and the membrane-binding sites of the toxin (in different color), as this would facilitate a clearer comparison between this map and the previously published X-ray structure.
15. 'Two hydrophilic residues (Y112 and Y148) were marked at the bending starting point, which might act as initial lipid interaction sites as well as maintain the inter-protomeric contact to form the channel (Figure S8E).' Should be Figure S9E.
16. Please double check the typo throughout the manuscript.

Reviewer #2 (Comments to the Authors (Required)):

Review Dutta JCB Aug 2025

In this manuscript the authors determined the structures of the α -hemolysin, a pore-forming toxin, in a series of conformations that convincingly represent the assembly of the oligomeric structure in the membrane. Although, the structure of α -hemolysin in the heptameric transmembrane state has been previously solved by x-ray crystallography, this manuscript solved the structure in a native lipid environment using cryo-EM methods. In addition, they solved several 'pre-pore' and smaller oligomers by lowering the temperature and decreasing the time of incubation of the toxin with the erythrocyte membranes. The group also examined the effects of adding the toxin to cells by fluorescent microscopy assays and confirmed membrane deformations and cell death. The authors also analyzed the lipid environment surrounding the toxin pore. The overall findings nicely define a credible pathway for α -hemolysin pore formation from 4-mer arc-like, to several intermediate pre-pores, to fully formed pore structures. Recently another manuscript, published by the same authors, explores the pre-pore α -hemolysin complex using synthetic liposomes that stabilize pre-pore complexes by modifying the lipid chain length (Chatterjee et al, Nat Commun, 2025). Including a direct comparison between the pre-pore structures from the two manuscripts would strengthen this manuscript by highlighting novel results and structural similarities to further support the author's pore formation assembly model.

Comments:

1. For non-experts in the field, an introduction to the structural features would be beneficial, with a model that labels the Cap, Rim, Stem and N-terminal domains.
2. Recommend expanding the discussion to include a comparison between other pre-pore and pore structures within the PFT family and clarify similarities and differences. Several previous structures are mentioned throughout the manuscript but without any discussion of structural differences. In addition, a comparison between the cryo-EM membrane-inserted model presented here to the previous detergent soluble x-ray model(s) would be beneficial, highlighting any differences, especially in the lipid headgroup binding sites. For example, discuss the binding of the phosphocholine head groups described in the Galdiero & Gouaux (2004) and MPD in Tanaka et al (2011) to the PC head group binding site shown in Fig. 3F. Are residues W179 and R200 in the RIM region, previously defined in Tanaka et al 2011, in the same location as shown here in the membrane bound form? Also, it is unclear if the structure of α -hemolysin in the membrane differs from previous detergent soluble structures or the recent structure solved in synthetic liposome (Chatterjee et al, Nat Commun, 2025).
3. For the pre-pore structures the low temperature (4 C) is concerning considering the erythrocyte membrane transition temperature is reported to be ~18-28 C. It is mentioned that the lower temperature makes the membrane more rigid and "Another possibility is that the rigid membrane did not allow the barrel to penetrate the bilayer and forced the transmembrane region to be distorted". With this as a possibility, further discussion of the pre-pore structures validity should be addressed.
4. The figures should highlight features mentioned in the text and the figure legends need to define all components in the figure (colored regions etc.). Specific examples area listed below under minor comments.
5. Examine the pre-pore structures without C7 symmetry applied to determine if the asymmetry exists in these structures. Also, is it standard practice to use a known structure (7AHL) as a reference for heterogeneous refinement and non-uniform refinement even if filtered to 40 Å?

Minor comments:

Define the phosphatidylcholine (fatty acid tails) used for Circular dichroism spectroscopy in methods.

Fig 1 - The images look over-saturated

Fig S4A - Show a high quality SDS-PAGE gel with all lanes labeled.

Line 247: "...a clear extra density was visible surrounding the TM segment and lower rim domain." Reference Fig 3C (right side,

gray).

Line 252: "Interestingly, a small population of pre-pore species also existed in this condition (Figure 3D; S4C)" It is not clear what 'condition' this sentence is referring to; barrel region not blocked by lipids or the prep in general?

Fig 4A - Include a direct comparison between high and low temp on one graph.

Sentence repeated on lines 317 and 321.

Fig 5A - in legend mention what is colored orange and highlight the "first few amino acid densities in the upper non-transmembrane segment of the barrel" line 336

Fig S7 - For comparison, show a fully formed pore as same threshold (0.024) illustrating an unblocked pore. There are no top/bottom views of the fully formed pore.

Fig S7C- Highlight the extra density corresponding to the transmembrane B-strand.

Lines 391: "...structural evidence that N-terminal region interacts with upper cap domain of the same and adjacent protomer." Can this be highlighted in one of the figures?

Label D108, T109, K110, Y148, V149, and Q150 residues in Fig 5J.

Highlight the segments shown in Fig S9 E&F in Fig S9A to quickly orient reader.

Highlight in figure - "...a small upward shift of the loop present in pre-pore state II."

No panels are labeled in S8

"Two hydrophilic residues (Y112 and Y148) were marked at the bending starting point, which might act as initial lipid interaction sites as well as maintain the inter-protomeric contact to form the channel (Figure S8E)."

Show the complete model of the pore structure presented in the manuscript, without density, in figure 3 and highlight any changes from the crystal structure 7AHL.

We want to thank the Editor for sending our manuscript for review and for sharing their valuable thoughts and suggestions about our manuscript. We would also like to thank all the reviewers for their time and effort in thoroughly reviewing our manuscript. We are grateful for their constructive suggestions and comments on our manuscript, which helped us to improve the manuscript. **For the last one month, we have performed negative staining TEM and cryo-EM imaging to validate the correlative imaging data, cryo-EM data collection, and 2D class averaging to improve the 2D class averages data quality of toxins incubated with HL-60 cells, and structural comparison analysis of our structures with previously known structures. We also performed biochemical experiments to validate our results. We have modified the figures at the request of the reviewers. Therefore, we hope we have successfully addressed all the reviewers' concerns in this current manuscript.** By addressing these comments, we feel the quality and impact of our manuscript have significantly improved. Nevertheless, these improvements and new data do not affect, modify, or contradict our previous conclusions. These have strengthened our hypothesis and allowed us to visualize the Stepwise Assembly of α -Hemolysin Leading to Erythrocyte Membrane Lysis. We hope that you will find the responses to the reviewers' comments satisfactory. Therefore, our current, improved manuscript is now suitable for publication as an article in the *Journal of Cell Biology*.

Point by point answer to the reviewers:

Reviewer #1 (Comments to the Authors (Required)):

Chatterjee et al. conducted a comprehensive functional and structural analysis of α -Hemolysin on erythrocyte membranes. They demonstrated that hemolysin oligomerizes upon incubation with HL-60 cells and erythrocytes. High-resolution structures of the mature pore form were obtained, revealing potential visualization of the phosphatidylcholine (PC) headgroup, along with a minor conformation representing the pre-pore state. To capture additional intermediate pre-pore stages, the authors lowered the incubation temperature, which allowed them to observe several transitional pre-pore conformations. These intermediates exhibit significant differences in the β -barrel region, suggesting a sequential transition among states. Additionally, pre-pore assemblies with 4-mer, 5-mer, and up to 8-mer configurations were identified, displaying varying degrees of openness (arc pre-pores), illustrating the progression from monomer binding to heptameric pre-pore assembly.

Overall, the work is well performed and described; the structural analysis is sound. The figures presentation should be improved, though. I have several comments:

Answer: Thank you for these encouraging comments and suggestions.

1. Why the pore-form and pre-pore form hemolysin oligomer are mostly detached from membrane in the cryo-EM sample?

Answer: Thank you for your query. This hemolysin toxin oligomerizes in the presence of phosphatidylcholine-containing lipid membranes. We performed all the experiments by incubating the toxins in the presence of cells, such as RBCs and HL-60, where a huge amount of phosphatidylcholine is available. The toxin molecules, upon binding to the cells, disrupt the plasma membrane, which is called the post-hemolytic condition. In this post-hemolytic condition, the plasma membrane was ruptured and therefore not clearly visible. Many regions of the image appeared as torn membrane fragments, showing clusters of lipid–protein mixtures, as highlighted in the figure below. Often, these ruptured lipid boundaries are not visible in cryo-EM micrographs due to their weak signal. However, they become much clearer when a low-pass filter is applied or when the images are denoised. However, a clear protein-lipid cluster was visible in the micrographs (Figure R1).

In a pre-hemolytic condition, the integrity of the plasma membrane bilayer was also highly compromised when the toxin was added. Therefore, the toxin-containing, compromised, and damaged lipid bilayer was not clearly visible. Furthermore, we have also shown the α -HL particles considered for structure determination, which are associated with a ruptured lipid bilayer (Figure R2). However, in some images, we observed heptameric oligomers in the background, and they appeared without any obvious ruptured lipid membranes. We believe that these heptameric oligomers are still surrounded by lipids, likely derived from the plasma membrane during rupture. We want to emphasize one important point about such proteins: the β -barrel and hydrophobic rim-domain regions of these toxins, which would be completely unstable in the absence of lipids or detergents, and would otherwise trigger protein aggregation. Thus, even the isolated toxins visible in the background are likely to be encapsulated in lipid moieties.

Figure R1. Representative cryo-EM micrographs of α -HL oligomers with RBCs in post-hemolytic condition. Left panel shows the raw micrographs. Middle panel

shows the boundaries of lipid-protein clusters in yellow color. The right panel shows the toxins (in green color) which were picked for structure determination.

Figure R2. Representative cryo-EM micrographs of α -HL oligomers with RBCs in pre-hemolytic condition. Left panel shows the raw micrographs. Middle panel shows the membrane portion of the RBCs in yellow color. The right panel shows the toxins (in green color) that were picked for structure determination.

2. Figure S1-C: 'The cryo-EM micrographs depicted the presence of circular toxins on the cell membrane.' Could the authors clarify whether the circular toxin structures observed are present on the vesicles or free in solution in the samples containing HL-60 cells?

Answer: Thank you for your comment. We were able to see the oligomeric toxins both on the intact vesicles as well as in the lipid-protein cluster form (where the plasma membrane was ruptured). The 2D class averages shown for oligomeric toxins in the presence of HL-60 cells (in Figure S1C) are picked from both areas. Some of the raw micrographs are shown here to depict the distribution of the toxins on the plasma-membrane-derived vesicles (Figure R3).

Figure R3. Representative cryo-EM micrographs of α -HL oligomers with extracellular vesicles of HL-60 cells.

As we mentioned in our previous answer, we want to clarify one important point about such kind of toxin proteins: the β -barrel and hydrophobic rim-domain regions of these toxins, which would be completely unstable and collapsed in the absence of lipids or detergents, and would otherwise trigger protein aggregation. Thus, even the isolated toxins visible in the background are likely surrounded by lipid moieties. Thus, in cryo-EM images, a few appear as lipid-free isolated toxins; however, even those isolated toxins are actually surrounded by lipids.

3. The 2D class averages are not convincing. Given the diversity of proteinous densities in the raw micrograph, it is unclear whether the 2D average is actually from the designated toxins. Please provide better 2D class averages and indicate the possible pore particles. Otherwise, the authors may consider to remove this 2D class averages.

Answer: Thank you for these suggestions. We agree with the reviewer that the previous 2D averages were not good and convincing. Therefore, we collected fresh data (877 micrographs) to improve the 2D classification and performed the entire data processing. We have improved the 2D class averages that were presented in Figure S1C.

Figure S1C. Cryo-EM data processing pipeline of α -HL in the presence of HL-60 cells. Raw micrograph of α -HL in the presence of HL-60 cells and 2D class averages.

4. The HL-60 derived vesicles do not have much toxins. Why? One suggestion would be to re-add an excessive amount of toxins to the isolated HL-60-derived vesicles and then check whether hemolysin decorates these vesicles.

Answer: Thank you for your query. Initially, we initially tried the experiment by adding a smaller amount of toxins, where we found less amount of toxins on plasma-membrane derived vesicles. As per your suggestions, we performed cryo-EM analysis with two more different toxin concentrations, i.e., 0.5mg/ml and 3mg/ml. In the first case (0.5mg/ml), we were able to see a good amount of toxins on the vesicle, and the vesicles were looking intact too (Figure R3). But the addition of excessive toxin (3mg/ml) resulted in complete rupture of the membrane and only lipid-protein-protein cluster, which is shown in Figure R4. Even our confocal microscopy study showed that the HL-60 cells are ruptured more rapidly than RBCs in the presence of α -HL. Altogether, these results suggested that HL-60 cells are delicate and more prone to toxin-mediated damage.

Figure R3. Representative cryo-EM micrographs of α -HL oligomers with extracellular vesicles of HL-60 cells.

Figure R4. Representative cryo-EM micrographs of α -HL oligomers with HL-60 cells when added in an excess amount, showing ruptured membrane and toxin clusters.

5. Figure 2B, please indicate the possible rings in the B(iii) and B(v). A higher magnification or zoom-in view will be helpful.

Answer: Thank you for pointing out this issue. We have marked the circular-shaped toxins using a yellow circle. These circular particles were picked and considered to perform 2D class averages from 50 micrographs (shown in Figure 1B, iv). In Figure 1B (v), we wanted to show the ruptured cells. The zoomed view of one part of the ruptured cells is represented to demonstrate the circular toxin on the membrane in Figure R5.

Figure R5. Different stages of RBC damage were captured using NS-TEM after adding the toxin i. untreated control RBC, ii. Damaged membrane of RBCs after α -

HL addition, iii. Zoomed view of membrane area showed circular-shaped particles, iv. 2D class averages of the particles confirmed the circular shape of the toxin, v. Lysed cell after completion of hemolysis. Zoomed view of figures iii and v showing the circular-shaped particles in yellow circles.

6. Figure 2E, the "correlative microscopy" data is not convincing, unless the authors statistically quantify the distortions and check whether the statistics in cryo-EM matches with fluorescent microscopy. The intact RBC and toxin-treated RBC are both too thick to be well-imaged by cryo-EM; the morphology changes under the cryo_EM conditions could be due to other physical changes during plunge freezing as well. The fluorescent imaging data is convincing and well presented to indicate the morphology change.

Answer: Thank you for the constructive comments and suggestions. This suggestion really helped us to revisit the methods and experiments. We do not have access to highly expensive CLEM equipment, which could have been implemented to study correlative microscopy data. However, in both cases (CLEM equipment and our strategy), the concept was the same: we visualized the samples using fluorescence microscopy and correlated the same samples using cryo-EM. We used the same samples with fluorescence microscopy and correlated the morphological changes of cells using TEM and cryo-EM, attempting to observe blebbing and other morphological variations (Figure R6-8). All our microscopy images correlated well with each other. However, we fully agree with the reviewer that "intact RBC and toxin-treated RBC are both too thick to be well-imaged by cryo-EM; the morphology changes under the cryo-EM conditions could be due to other physical changes during plunge freezing as well." Thus, we repeated the same experiments using confocal and room-temperature TEM, where blotting force (physical changes during plunge freezing) would not affect cell morphology (Figure R7-8). The results were consistent, and we observed similar morphological changes as seen in cryo-EM and confocal microscopy. We also collected ~50 images at lower magnification from different grids to statistically quantify the distortions. The best correlating images were taken to further compare the distortion in cryo-EM images with fluorescence images (Figure R6A).

Furthermore, our results demonstrated that the outer membrane became blurred in toxin-treated RBCs when imaged under both NS-TEM and cryo-EM (Figure R6, R8). The high-magnification image in cryo-EM showed ruptured membranes in the post-hemolytic condition (Figure R6E). We have incorporated these images in the supplemental figure Figure S2, which will help the reader to understand the comparison properly.

Additionally, we have noticed similar approaches were implemented in previous literatures. The references are listed, where authors have done whole cell imaging for in-situ structural cell biology using cryo-EM (rebuttal references 1-3).

A. Correlative imaging toxin-RBCs under confocal and cryo-EM

B. Control RBC

C. RBC in pre-hemolytic condition

Zoomed view

D. RBC in post-hemolytic condition

E. Cryo-EM micrographs of α -HL with RBC in post-hemolytic condition

Zoomed view

F. Cryo-EM micrographs of α -HL with RBC in pre-hemolytic condition

Figure R6. Cryo-EM analysis of pre-hemolytic and post-hemolytic states of RBCs.

A. Correlation of toxin-mediated damage in RBCs using high-resolution cryo-EM imaging with the confocal data (shown in top left). The size of the protruded portions from the cells under confocal and cryo-EM in the left side panel was around $1\mu\text{m}$. The ratio of width to length of cells under confocal and cryo-EM was 1.2 in the middle panel. The ratio of width to length of cells under confocal and cryo-EM was 0.74 in the right-side panel. **B.** Control RBC (entire cell). **C.** RBC (entire cell) in pre-hemolytic condition, and its zoomed image. **D.** RBC (entire cell) in post-hemolytic condition, and its zoomed view. **E-F.** Cryo-EM micrographs of toxin-containing membranes, either in lysed form or vesicular form, outside the RBCs in post- and pre-hemolytic conditions, respectively (scale bar 100 nm).

Figure R7. NS-TEM images of control RBCs at different magnifications.

Figure R8. NS-TEM images of toxin-treated RBCs at different magnifications.

Rebuttal References:

1. Sartori-Rupp, A., Cordero Cervantes, D., Pepe, A., Gousset, K., Delage, E., Corroyer-Dulmont, S., ... & Zurzolo, C. (2019). Correlative cryo-electron microscopy reveals the structure of TNTs in neuronal cells. *Nature communications*, 10(1), 342.
2. Tivol, W. F., Briegel, A., & Jensen, G. J. (2008). An improved cryogen for plunge freezing. *Microscopy and Microanalysis*, 14(5), 375-379.
3. Sibert, B. S., Kim, J. Y., Yang, J. E., & Wright, E. R. (2021). Whole-cell cryo-electron tomography of cultured and primary eukaryotic cells on micropatterned TEM grids. *bioRxiv*, 2021-06.

7. L254- prepore state along the membrane? Or also in the solution? In Figure 4D, there are particles in both solution state and membrane bound state. Are the membrane-bound structure in a compositional and conformational different state from solution oligomers?

Answer: Thank you for this query. We have observed that sometimes the lipid bilayer is properly visible and sometimes it is not, after the disruption caused by the toxin molecules. We have highlighted the lipid layers/plasma membrane layers to show the membrane-mediated oligomerization. When we backtraced the particles used in solving the structure, we noticed that particles were picked from those highlighted areas, where the lipid layers are also visible (in some cases very prominent and in some cases not very prominent) (Figure R2). Also, as mentioned earlier, oligomer (pore and/or pre-pore complex) can't exist without lipids. So whichever molecules appear isolated are either embedded in ruptured membranes or isolated lipid moieties surrounding the rim and β -barrel region.

Figure R2. Representative cryo-EM micrographs of α -HL oligomers with RBCs in pre-hemolytic condition. Left panel showing the raw micrographs. Middle panel showing the membrane portion of the RBCs in yellow color. The right panel showing the toxins (in green color) which were picked up for structure determination.

8. L266, the presence of both unsaturated and saturated lipids with different chain lengths, leads to instable membrane bilayer? Any reference for this?

Answer: Apology for the confusion. Here, we wanted to mean “unstable” equivalent to non-rigid/non-compact. Therefore, we have changed the word “unstable” to “flexible” in line no. 267. Most of the references also highlighted that the lipid bilayer is flexible if it is made by a combination of saturated and unsaturated lipids (rebuttal references 4-5). Thus, we also change the word unstable to “flexible”.

Rebuttal References:

4. *Martinez-Seara, H., Róg, T., Pasenkiewicz-Gierula, M., Vattulainen, I., Karttunen, M., & Reigada, R. (2008). Interplay of unsaturated phospholipids and cholesterol in membranes: effect of the double-bond position. Biophysical journal, 95(7), 3295-3305.*

5. *Leekumjorn, S., Cho, H. J., Wu, Y., Wright, N. T., Sum, A. K., & Chan, C. (2009). The role of fatty acid unsaturation in minimizing biophysical changes on the structure and local effects of bilayer membranes. Biochimica et Biophysica Acta (BBA)-Biomembranes, 1788(7), 1508-1516.*

9. Figure S5 resolution distribution plot should be better presented. The color is confusing.

Answer: Thank you for your suggestions. We have presented the local resolution estimation of the high-resolution maps with a better set of colors, which can be easily distinguished (Figure S4).

Figure S4. Local resolution estimation in the half maps of different pre-pore conformations.

10. In Figure S5, the pre-pore form without C7 symmetry should be performed as well. Any twist in structures?

Answer: Thank you for your query and suggestion. Here, we have shown the pre-pore structures after processing without imposing any symmetry. From the map-to-model fitting, we could not distinguish any significant changes in the structures when compared to structures processed with C7 symmetry (Figure R9-R11). Also, we have incorporated these cryo-EM maps in supplemental Figure S4, where we have demonstrated the data processing pipeline.

A. Cryo-EM structure of α -HL pre-pore state II (C1 symmetry)

map

Map to model fitting

One protomer fitting

B. Cryo-EM structure of α -HL pre-pore state II (C7 symmetry)

map

Map to model fitting

One protomer fitting

Figure R9. Cryo-EM analysis of α -HL heptameric pre-pore state II with **A.** C1 symmetry, and **B.** C7 symmetry.

A. Cryo-EM structure of α -HL pre-pore state III (C1 symmetry)

map

Map to model fitting

One protomer fitting

B. Cryo-EM structure of α -HL pre-pore state III (C7 symmetry)

map

Map to model fitting

One protomer fitting

Figure R10. Cryo-EM analysis of α -HL heptameric pre-pore state III with **A.** C1 symmetry, and **B.** C7 symmetry.

Figure R11. Cryo-EM analysis of α -HL heptameric pre-pore state IV with **A.** C1 symmetry, and **B.** C7 symmetry.

11. Have the authors attempted incubating HL-60 cells with α -hemolysin at 4°C, similar to the conditions used with RBCs? Such an approach could help capture the pre-pore state or yield supernatant with less intracellular material, thereby reducing background noise in micrographs.

Answer: Thank you for your suggestions. As per your suggestions, we have performed the experiments of α -hemolysin with HL-60 cells at 4 °C. We collected a dataset of 602 micrographs to get an idea about different oligomeric species by doing 2D classification. The 2D class averages showed the presence of some arc-like species apart from heptameric species (Figure R12). As the particle number was coming low after rigorous 2D classification, we did not perform any 3D reconstruction. However, we did not try to collect more data and reconstruct the map because we didn't see any significant difference between 4 °C RBC and HL60 cells. Additionally, this would have complicated the story to include both 4 °C RBC and HL60 cells, where overall architecture appeared almost identical. Therefore, we have not incorporated the data into the main manuscript. We consider this an excellent suggestion, and we are further extending our studies with HL-60 and macrophage cells. We will employ the same strategy in our new study and compare the structural difference between α -HL at 4 °C and with immune cells, like macrophages and neutrophils.

Figure R12. Cryo-EM analysis of α -HL in the presence of HL-60 cells incubated at 4°C.

12. α -Hemolysin (α -HL) is known to form pores not only in leukocytes and erythrocytes but also in other cell types such as epithelial and endothelial cells. Have the authors investigated the pore-forming activity of α -HL in these additional cell types?

Answer: Thank you for your query. We agree with the reviewer. It is well known that α -HL affects various epithelial and endothelial cells (rebuttal references 6-8). However, we did not directly test α -Hemolysin (α -HL) activity on epithelial or endothelial cells in this study. We have been working on the structural and functional characterization of α -Hemolysin (α -HL) and bi-component toxins on some other cell types in a separate study, and we did not incorporate those data here, as it would make the current work more complicated. Moreover, previous studies have already reported α -Hemolysin (α -HL) activity on epithelial and endothelial cells (rebuttal references 6-8).

Rebuttal References:

6. Kwak, Y. K., Vikström, E., Magnusson, K. E., Vécsey-Semjén, B., Colque-Navarro, P., & Möllby, R. (2012). *The Staphylococcus aureus alpha-toxin perturbs the barrier function in Caco-2 epithelial cell monolayers by altering junctional integrity. Infection and immunity, 80(5), 1670-1680.*

7. Becker, K. A., Fahsel, B., Kemper, H., Mayeres, J., Li, C., Wilker, B., ... & Gulbins, E. (2018). *Staphylococcus aureus alpha-toxin disrupts endothelial-cell tight junctions via acid sphingomyelinase and ceramide. Infection and immunity, 86(1), 10-1128.*

8. Yang, C., Robledo-Avila, F. H., Partida-Sanchez, S., & Montgomery, C. P. (2024). *α -Hemolysin-mediated endothelial injury contributes to the development of Staphylococcus aureus-induced dermonecrosis. Infection and immunity, 92(8), e00133-24.*

13. Figure 3 C: It would be beneficial to include a label indicating the size of the pore.

Answer: Thank you for your suggestions. We have labelled the pore size at the top of the cap domain and the bottom of the barrel region (stem domain) in Figure 2C. The sizes of the pores in these two regions are 2.68 nm and 2.57 nm, respectively.

Figure 2C. Cryo-EM map of α -HL pore form in the presence of RBC in post-hemolytic condition.

14. Figure S4C: 'a clear extra density was visible surrounding the TM segment and lower rim domain. We speculated that density was coming from the membrane components and was found to be nicely decorating the membrane binding sites of the toxin.' It would be helpful to highlight the regions of additional density and the membrane-binding sites of the toxin (in different color), as this would facilitate a clearer comparison between this map and the previously published X-ray structure.

Answer: Thank you for your suggestions. We have shown the extra lipid density. The fitted atomic model shows the extra lipidic part. Additionally, the fitted C₁₄PC lipid head group in our RBC-derived pore structure demonstrated that the location of the lipid head group has extra density in our pore structure. Apart from that, the lipid density covers the entire transmembrane region and the lower part of the rim domain (Figure R13). The extra density in the map is also highlighted in Figure S3C.

A. Fitting of the lipid head group containing crystal structure into the pore structure

Figure R13A. Cryo-EM map of pore form showing extra lipid density (Yellow color), crystal structure (PDB: 6U49) fitted map, zoomed view of the fitted map showing lipid density covering the lipid head groups present in the crystal structure.

15. 'Two hydrophilic residues (Y112 and Y148) were marked at the bending starting point, which might act as initial lipid interaction sites as well as maintain the inter-protomeric contact to form the channel (Figure S8E).' Should be Figure S9E.

Answer: Thank you for pointing out the mistake. We have changed the figure number to Figure S9E. However, currently this is shown in Figure 6E.

16. Please double check the typo throughout the manuscript.

Answer: Thank you for pointing out the issue. We have checked the typo throughout the manuscript and made the changes at the appropriate positions. The changes are listed below:

Abstract:

“different toxin conformations, identified” to “different conformations of α -HL, identified”

Introduction:

“This is the most common bacteria that cause” to “This is the most common bacteria that causes”

“in an increase in evaluating and understating the pathophysiology” to “extremely challenging to combat *S. aureus* (Chambers and DeLeo, 2009). To address this issue, it is crucial to study the pathophysiology of *S. aureus*”.

“cells shedding the microvesicles, helped in toxin clearance” to “where the cells shed microvesicles, helping in toxin clearance”.

Results:

“the intensity of DAPI got significantly reduced” to “the intensity of DAPI was significantly reduced”

“This might be possible because of the nucleus of the HL-60 cells got disrupted ” to “that due to membrane rupture, the nucleus of the HL-60 cells was disrupted”

“This necrosis-mediated cell death process happens” to “This necrosis-mediated cell death process happened”

“Further structural analysis were performed” to “Further structural analysis was performed”

“the transmembrane segment were mixed” to “the transmembrane segment was mixed”

“identify lipid moieties attached with the toxin molecules” to “identify lipid moieties attached to the toxin molecules”

“We speculated two methods” to “We considered two methods”

“This indicated the compact packing of the RBC” to “This indicated that the compact packing of the lipid components of the RBC plasma membrane”

“So far, our cryo-EM results demonstrated” to “So far, our cryo-EM results have demonstrated”

“the presence of the adjacent protomer is very crucial” to “the presence of the adjacent protomer was very crucial”

“N-terminal region interacts with upper cap domain” to “N-terminal region interacted with upper cap domain”

“where pre-stem was released” to “where pre-stem was released”

Discussion:

“plasma membrane in pore formation are elucidated” to “plasma membrane (heterogeneous lipid environment) in pore formation was elucidated”

“oligomerization process lead to the generation” to “oligomerization process led to the generation”

Reviewer #2 (Comments to the Authors (Required)):
Review Dutta JCB Aug 2025

In this manuscript the authors determined the structures of the α -hemolysin, a pore-forming toxin, in a series of conformations that convincingly represent the assembly of the oligomeric structure in the membrane. Although, the structure of α -hemolysin in the heptameric transmembrane state has been previously solved by x-ray crystallography, this manuscript solved the structure in a native lipid environment using cryo-EM methods. In addition, they solved several 'pre-pore' and smaller oligomers by lowering the temperature and decreasing the time of incubation of the toxin with the erythrocyte membranes. The group also examined the effects of adding the toxin to cells by fluorescent microscopy assays and confirmed membrane deformations and cell death. The authors also analyzed the lipid environment surrounding the toxin pore. The overall findings nicely define a credible pathway for α -hemolysin pore formation from 4-mer arc-like, to several intermediate pre-pores, to fully formed pore structures. Recently another manuscript, published by the same authors, explores the pre-pore α -hemolysin complex using synthetic liposomes that stabilize pre-pore complexes by modifying the lipid chain length (Chatterjee et al, Nat Commun, 2025). Including a direct comparison between the pre-pore structures from the two manuscripts would strengthen this manuscript by highlighting novel results and structural similarities to further support the author's pore formation assembly model.

Answer: Thank you for these encouraging comments. We have incorporated a comparison study in this manuscript.

Comments:

1. For non-experts in the field, an introduction to the structural features would be beneficial, with a model that labels the Cap, Rim, Stem and N-terminal domains.

Answer: Thank you for your suggestions. In our revised figures, we have labeled the cap, rim, stem, and N-terminal in cryo-EM maps as well as in the atomic models.

Figure 2C. Cryo-EM maps of α -HL heptameric pore structure at 3.1 Å resolution, where each protomer is shown in different colors (from left, lime green, golden rod, cyan, hot pink, olive drab, dodger blue, medium orchid). The cross-section of the map showed a pore channel and extra density (gray color) corresponding to the membrane surrounding the barrel and the lower part of the rim domain.

Figure S3D. Atomic model of the heptameric pore structure (in cyan).

2. Recommend expanding the discussion to include a comparison between other pre-pore and pore structures within the PFT family and clarify similarities and differences. Several previous structures are mentioned throughout the manuscript but without any discussion of structural differences. In addition, a comparison

between the cryo-EM membrane-inserted model presented here to the previous detergent soluble x-ray model(s) would be beneficial, highlighting any differences, especially in the lipid headgroup binding sites. For example, discuss the binding of the phosphocholine head groups described in the Galdiero & Gouaux (2004) and MPD in Tanaka et al (2011) to the PC head group binding site shown in Fig. 3F. Are residues W179 and R200 in the RIM region, previously defined in Tanaka et al 2011, in the same location as shown here in the membrane bound form? Also, it is unclear if the structure of α -hemolysin in the membrane differs from previous detergent soluble structures or the recent structure solved in synthetic liposome (Chatterjee et al, Nat Commun, 2025).

Answer: Thank you for your suggestions. As per your suggestions, we have shown a detailed comparison of pore structure solved from RBCs with detergent-solubilized and liposome-based stabilized pore structures (Figure 3). The RMSD of pore structure from RBCs with MPD solubilized pore (PDB:3ANZ), pore crystal structure with PC-head group (PDB: 6U49), 10:0 PC-liposome derived pore structure (PDB:9KRF), and detergent-solubilized pore structure (PDB: 7AHL) are 0.548 Å, 0.560 Å, 0.152 Å, and 0.532 Å, respectively. These comparative studies demonstrate that the α -HL toxin's overall structures in the cellular environment, particularly the cap region, are highly similar to the detergent-solubilized crystal structure of α -HL. However, subtle differences were observed in the rim and stem regions, which directly interact with the lipid membrane. Notably, a shift was detected in the rim domain loops (at residues S262 and G180, displaced by 2.4 Å and ~2 Å, respectively) when our structure was compared with the crystal structures. Interestingly, these residues correspond to lipid-binding amino acids. In contrast, the RMSD remained very low compared to the liposome-derived cryo-EM structure. Thus, we were able to identify minor structural differences between the detergent-solubilized crystal structure and the lipid-binding α -HL structure reported in this manuscript. Furthermore, the lipid head-group binding site calculated from our structure aligns well with the crystal structure containing MPD and a lipid head group. Importantly, our current study also revealed multiple intermediate conformations of these oligomeric toxins, which cannot be captured by detergent-solubilized crystal structures; therefore, highlighting the unique advantage of cellular environments in resolving functionally relevant states of these toxins.

Figure 3. Comparison of lipid binding sites in different detergent-solubilized and liposome-based stabilized pore structures. **A.** Comparison of pore structure in the presence of RBCs (cyan) with pore as well as in the presence of MPD (yellow). The zoomed view shows the location of the lipid head group and MPD density. **B.** Comparison of pore structure solved in the presence of RBCs (cyan) with the pore structure solved in the presence of 10:0 PC liposomes (orange). The zoomed view shows the location of the lipid head group and 10:0 PC density. **C.** Comparison of pore structure solved in the presence of RBCs (cyan) with the pore structure solved in the presence of C₁₄ PC (green). **D.** Comparison of pore structure solved in the presence of RBCs (cyan) with the pore structure solved in the presence of detergent (pink). The shifts in S262 and G180 residues in the rim domain are 2.4 Å and ~2 Å, respectively.

3. For the pre-pore structures the low temperature (4 C) is concerning considering the erythrocyte membrane transition temperature is reported to be ~18-28 C. It is mentioned that the lower temperature makes the membrane more rigid and "Another possibility is that the rigid membrane did not allow the barrel to penetrate the bilayer and forced the transmembrane region to be distorted". With this as a possibility, further discussion of the pre-pore structures validity should be addressed.

Answer: We completely agree with the reviewer. Here, we wanted to rigidify the plasma membrane to capture some intermediate species of α -HL, and therefore, we

used cold temperature to rigidify the lipid membrane. Similarly, we tried to rigidify the membrane in our liposome-based α -HL structure by introducing sphingomyelin into PC-containing liposomes (Chatterjee et al. 2025, Nat. Comm). We were able to resolve only heptameric pre-pores in the presence of sphingomyelin into PC-containing liposomes. Whereas in this case (pre-hemolytic condition and cellular environment), we were able to determine the non-TMs pre-pore (pre-pore II) and late pre-pore (pre-pore IV) structure, as well as resolve the pre-pore state III, where the barrel was bent. If we consider the membrane to be completely rigid, we would have ended up with only pre-pore state II, where TMs are absent. However, we resolved two other states where a partial TMs is present. Therefore, we speculated that these are also pre-pore species where the β -barrel is distorted/disordered, and one reason could be that the membrane is rigid. In this way, we were able to demonstrate some heptameric pre-pore intermediate species in cellular conditions. MD simulation work in a membrane environment can further validate this intermediate species, which we might explore in the near future. Additionally, we have provided a comparison between pre-pore state II with pre-pore conformation from liposomes. In both cases, TMs were absent, but there is a structural difference with an RMSD of 1.1 Å. However, in this current pre-pore state II structure, we were able to resolve a few more non-TMs residues (Figure R14A). The comparison study of pre-pore state IV with late-pre pore structure from liposomes showed almost similar structures, with an RMSD of 0.17 Å (Figure R14B).

Figure R14A. Structural comparison of pre-pore state II with pre-pore conformation solved in the presence of liposome. **B.** Structural comparison of pre-pore state IV with late pre-pore conformation solved in the presence of liposome.

4. The figures should highlight features mentioned in the text and the figure legends need to define all components in the figure (colored regions etc.). Specific examples area listed below under minor comments.

Answer: Thank you for your suggestions. We have mentioned the colors in the figure legends wherever needed. The changes are listed below:

Figure 1:

The intensity of the nucleus (in blue color) staining with DAPI dye

Nile red intensity ($\lambda_{excitation}= 540$ nm, $\lambda_{emission}= 565-610$ nm) corresponding to the membrane (in red color)

Toxin treatment showed membrane protrusion (in red color)

The cell membranes (in red color) were stained

Zoomed view of the membrane area showed circular-shaped particles (yellow encircled),

A zoomed view of the lysed-membrane area showed circular-shaped particles (yellow encircled).

Figure 5:

Cryo-EM map of pre-pore state II, density corresponds to one protomer shown in golden rod color, and the rest of the protomers in gray color.

The D108, T109, K110, Y148, V149, and Q150 residues are shown in deep green color. The minor upward shift on some loops is highlighted using a black arrow.

Comparison between pre-pore state III (blue) vs state IV (deep pink).

Supplementary Figure 1:

The cell membranes (in red color) were stained using Nile Red dye ($\lambda_{excitation}= 540$ nm, $\lambda_{emission}= 565-610$ nm).

5. Examine the pre-pore structures without C7 symmetry applied to determine if the asymmetry exists in these structures. Also, is it standard practice to use a known structure (7AHL) as a reference for heterogeneous refinement and non-uniform refinement even if filtered to 40 Å?

Answer: Thank you for your query. Here, we have shown the pre-pore structures after processing without imposing any symmetry. From the map-to-model fitting, we could not distinguish any significant changes in the structures when compared to structures processed with C7 symmetry (Figure R9-R11). The maps processed in C1 symmetry are provided in Figure S4.

Additionally, it is a standard practice to use a low-pass filtered (low-resolution) map or a low-resolution map generated from a crystal structure to use as an initial model to avoid any model bias. Here, we low-pass filtered to 40 Å, where it looks like a blob, which basically helped us to get different conformational states and align the particles at the initial processing state (shown in Figures S3 and S4). Some reference literatures where a 40 Å low-pass filter was applied to the reference maps are listed below (rebuttal references 9-12).

Rebuttal references:

9. Lin, X., Chen, B., Wu, Y., Han, Y., Qi, A., Wang, J., ... & Xu, F. (2023). Cryo-EM structures of orphan GPR21 signaling complexes. *Nature communications*, 14(1), 216.

10. Nakane, T., Kimanius, D., Lindahl, E., & Scheres, S. H. (2018). Characterisation of molecular motions in cryo-EM single-particle data by multi-body refinement in RELION. *elife*, 7, e36861.

11. Ilangovan, A., Kay, C. W., Roier, S., El Mkami, H., Salvadori, E., Zechner, E. L., ... & Waksman, G. (2017). Cryo-EM structure of a relaxase reveals the molecular basis of DNA unwinding during bacterial conjugation. *Cell*, 169(4), 708-721.

12. Tan, Y. Z., Baldwin, P. R., Davis, J. H., Williamson, J. R., Potter, C. S., Carragher, B., & Lyumkis, D. (2017). Addressing preferred specimen orientation in single-particle cryo-EM through tilting. *Nature methods*, 14(8), 793-796.

Figure R9. Cryo-EM analysis of α -HL heptameric pre-pore state II with **A.** C1 symmetry, and **B.** C7 symmetry.

A. Cryo-EM structure of α -HL pre-pore state III (C1 symmetry)

map

Map to model fitting

One protomer fitting

B. Cryo-EM structure of α -HL pre-pore state III (C7 symmetry)

map

Map to model fitting

One protomer fitting

Figure R10. Cryo-EM analysis of α -HL heptameric pre-pore state III with **A.** C1 symmetry, and **B.** C7 symmetry.

Figure R11. Cryo-EM analysis of α -HL heptameric pre-pore state IV with **A.** C1 symmetry, and **B.** C7 symmetry.

Minor

comments:

Question: Define the phosphatidylcholine (fatty acid tails) used for Circular dichroism spectroscopy in methods.

Answer: Thank you for your suggestions. The modified sentence in lines 614-617 reads like “The melting studies of the toxin were performed with large unilamellar vesicles (LUVs), which were made up of 1-Palmitoyl-2-oleoyl-sn-glycero-3-phosphatidylcholine (POPC). POPCs stored at different temperatures (37°C and 4°C) were selected for performing Circular Dichroism spectroscopy using a JASCO J-715 Spectrophotometer.”

Question: Fig 1 - The images look over-saturated

Answer: Thank you for your suggestions. We have adjusted the brightness and contrast of this image. The background of these images is black, suggesting that the fluorescence is corresponding to the membrane.

Figure 1A. The intensity of nucleus staining with DAPI dye ($\lambda_{excitation} = 405 \text{ nm}$, $\lambda_{emission} = 480\text{-}510 \text{ nm}$) increased for the first couple of minutes, where the three nuclear lobes (white arrowhead) became clearly visible due to the uptake of the dye from the outside medium. After some time (5 minutes), the intensity of the nucleus started decreasing, and three distinct lobes of the nucleus started disappearing. Nile red intensity ($\lambda_{excitation} = 540 \text{ nm}$, $\lambda_{emission} = 565\text{-}610 \text{ nm}$) started decreasing rapidly just after toxin treatment at a 100 nM toxin concentration. **B.** Toxin treatment showed membrane protrusion from HL-60 cells at 100 nM toxin concentration.

Question: Fig S4A - Show a high quality SDS-PAGE gel with all lanes labeled.

Answer: Thank you for your generous advice. We have added the high-quality SDS-PAGE gel in Figure S3A.

Figure S3A. Sample preparation procedure of α -HL-treated RBCs (Created with BioRender.com). SDS-PAGE gel showing lanes corresponding to untreated RBC pellet-lane 1, untreated RBC supernatant-lane 2, toxin-treated RBC pellet-lane 3, toxin-treated RBC supernatant-lane 4, and protein marker-lane 5.

Question: Line 247: "...a clear extra density was visible surrounding the TM segment and lower rim domain." Reference Fig 3C (right side, gray).

Answer: Thank you for pointing out this issue. We have added the reference to Figure 3C. The updated sentence in lines 249-250 is "Whereas, at a lower threshold, a clear extra density was visible surrounding the TM segment and lower rim domain (**Figure 2C**; **S3C-D**)."

Question: Line 252: "Interestingly, a small population of pre-pore species also existed in this condition (Figure 3D; S4C)" It is not clear what 'condition' this sentence is referring to; barrel region not blocked by lipids or the prep in general?

Answer: Thank you for your suggestions. The modified sentence, in lines 252-254, reads like "Interestingly, a small population of pre-pore species also existed in this condition, where β -barrel was not formed (**Figure 2D**; **S3C**)".

Question: Fig 4A - Include a direct comparison between high and low temp on one graph.

Answer: Thank you for your suggestions. We have incorporated the comparative hemolysis graph at high and low temperatures in Figure 4A.

Figure 4A. Compromised hemolysis of RBC at a lower temperature (10°C) with different toxin concentrations as compared to hemolysis at 37°C.

Question: Sentence repeated on lines 317 and 321.

Answer: Sorry for the mistake. We have removed the duplicate sentence corresponding to lines 335-337.

Question: Fig 5A - in legend mention what is colored orange and highlight the "first few amino acid densities in the upper non-transmembrane segment of the barrel" line 336

Answer: Sorry for the confusion. In Figure 5A, one protomer is colored in golden rod. We have mentioned in the figure legends. We have labelled the first few amino acid densities in the upper non-transmembrane segment of the barrel of pre-pore state II. Additionally, we have labeled the first few amino acids in Figures 5J and 6F.

Figure 5A. Cryo-EM map of pre-pore state II, density corresponds to one protomer shown in golden rod color, and the rest of the protomers in gray color.

Figure 5J. D108, T109, K110, Y148, V149, and Q150 residues are shown in deep green color.

Question: Fig S7 - For comparison, show a fully formed pore as same threshold (0.024) illustrating an unblocked pore. There are no top/bottom views of the fully formed pore.

Answer: Thank you for your suggestions. We have shown the α -HL pore state at different thresholds, even at a very high threshold, where dust starts coming, the pore is not blocked by lipids (Figure S6E).

Figure S6E. Cryo-EM map of α -HL pore state at different thresholds.

Question: Fig S7C- Highlight the extra density corresponding to the transmembrane B-strand.

Answer: Thank you for your suggestions. We have highlighted the extra density corresponding to the transmembrane β -strand in pre-pore state III in Figure S6C (previously it was Figure S7C) using a black arrow. A similar extra density is shown in Figure 5C-D. We have also referred to this Figure in the main text to indicate the extra density.

Figure S6C. Transmembrane β -barrel and lipid densities in pre-pore state III. Black arrowhead showing the extra β -strand density.

Figure 5C. Cryo-EM map of pre-pore state III, one protomer is shown in blue. **D.** Fitted atomic model of one protomer of pre-pore state III.

Question: Lines 391: "...structural evidence that N-terminal region interacts with upper cap domain of the same and adjacent protomer." Can this be highlighted in one of the figures?

Answer: Thank you for your suggestions. We have introduced this segment in Figure S5C and added the reference in the main text (line 402).

Figure S5C. Orientation of the pentameric structure. The red-box region shows the zoomed view of the N-terminal region. In the zoomed view, the N-terminal of one of the protomers is highlighted in golden rod color, showing its interaction with the upper cap domain.

Question: Label D108, T109, K110, Y148, V149, and Q150 residues in Fig 5J. Highlight the segments shown in Fig S9 E&F in Fig S9A to quickly orient reader.

Answer: Thank you for your suggestions. We have labeled these residues in Figure 5J.

Question: Highlight in figure - "...a small upward shift of the loop present in pre-pore state II."

Answer: Thank you for your valuable suggestions. We have highlighted the shift in Figure 5J.

Question: No panels are labeled in S8 "Two hydrophilic residues (Y112 and Y148) were marked at the bending starting point, which might act as initial lipid interaction sites as well as maintain the inter-protomeric contact to form the channel (Figure S8E)."

Answer: Apology for the mistake. It should have been labelled as "Figure S9E". However, currently this is shown in Figure 6F.

Question: Show the complete model of the pore structure presented in the manuscript, without density, in figure 3 and highlight any changes from the crystal structure 7AHL.

Answer: Thank you for your suggestions. We have incorporated the atomic model of pore structure in Figure S3D. The RMSD between these two structures is 0.532 Å. The comparison with the crystal structure (PDB ID: 7AHL) is represented in Figure 3D. The comparison showed the shift in the loops of the rim domain, which is the lipid-binding region.

Figure S3D. Atomic model of the heptameric pore structure from RBCs (in cyan).

Figure 3D. Comparison of the pore structure (in cyan) with previously solved detergent-solubilized crystal structure (PDB ID: 7AHL; in pink)

November 17, 2025

RE: JCB Manuscript #202506129R

Somnath Dutta
Indian Institute of Science Bangalore

Dear Prof. Dutta,

Thank you for submitting your revised manuscript entitled "Cryo-EM Captures the Stepwise Assembly of α -Hemolysin Leading to Erythrocyte Membrane Lysis." We would be happy to publish your paper in JCB pending final revisions necessary to address the remaining reviewer comments and meet our formatting guidelines (see details below). It would be important to address Reviewer #2's comments and clarify the novelty of the findings and advance in this work compared with your previous paper.

A. MANUSCRIPT ORGANIZATION AND FORMATTING:

1) Text limits: Character count for Articles is < 40,000, not including spaces. Count includes title page, abstract, introduction, results, discussion, and acknowledgments. Count does not include materials and methods, figure legends, references, tables, or supplemental legends.

2) Figure formatting: Articles may have up to 10 main text figures. Scale bars must be present on all microscopy images, including inset magnifications. Molecular weight or nucleic acid size markers must be included on all gel electrophoresis. Please add scale bars to figures 1E(iv) and magnifications in 2A & S3B.

Also, please avoid pairing red and green for images and graphs to ensure legibility for color-blind readers. If red and green are paired for images, please ensure that the particular red and green hues used in micrographs are distinctive with any of the colorblind types. If not, please modify colors accordingly or provide separate images of the individual channels.

3) Statistical analysis: Error bars on graphic representations of numerical data must be clearly described in the figure legend. The number of independent data points (n) represented in a graph must be indicated in the legend. Please indicate whether 'n' refers to technical or biological replicates (i.e. number of analyzed cells, samples or animals, number of independent experiments). If independent experiments with multiple biological replicates have been performed, we recommend using distribution-reproducibility SuperPlots (please see Lord et al., JCB 2020) to better display the distribution of the entire dataset, and report statistics (such as means, error bars, and P values) that address the reproducibility of the findings.

Statistical methods should be explained in full in the materials and methods. For figures presenting pooled data the statistical measure should be defined in the figure legends. Please also be sure to indicate the statistical tests used in each of your experiments (both in the figure legend itself and in a separate methods section) as well as the parameters of the test (for example, if you ran a t-test, please indicate if it was one- or two-sided, etc.). Also, if you used parametric tests, please indicate if the data distribution was tested for normality (and if so, how). If not, you must state something to the effect that "Data distribution was assumed to be normal but this was not formally tested."

4) Title: While your current title will be appreciated by the specialists, we do not feel that it will be accessible to a broader cell biology audience. Therefore to convey the major advance more clearly, we suggest the following title:
"Stepwise assembly of α -Hemolysin from a 5-mer to a complete octameric pore in native erythrocytes"

5) Materials and methods: Should be comprehensive and not simply reference a previous publication for details on how an experiment was performed. Please provide full descriptions (at least in brief) in the text for readers who may not have access to referenced manuscripts. The text should not refer to methods "...as previously described." Please make sure to provide a full description of the SDS-PAGE and immunoblotting procedures including the type of gel & membrane, blocking reagents, antibodies, and acquisition and quantification methods. Centrifugation speeds should be given in rcf/xg units and not rpm.

6) For all cell lines, vectors, strains, constructs/cDNAs, etc. - all genetic material: please include database / vendor ID (e.g. Addgene, ATCC, etc.) or if unavailable, please briefly describe their basic genetic features, even if described in other published work or gifted to you by other investigators (and provide references where appropriate). Please be sure to provide the sequences for all of your oligos: primers, si/shRNA, RNAi, gRNAs, etc. in the materials and methods. You must also indicate in the methods the source, species, and catalog numbers/vendor identifiers (where appropriate) for all of your antibodies, including

secondary. If antibodies are not commercial, please add a reference citation if possible.

7) Microscope image acquisition: The following information must be provided about the acquisition and processing of images:

- a. Make and model of microscope
- b. Type, magnification, and numerical aperture of the objective lenses
- c. Temperature
- d. Imaging medium
- e. Fluorochromes
- f. Camera make and model
- g. Acquisition software
- h. Any software used for image processing subsequent to data acquisition. Please include details and types of operations involved (e.g., type of deconvolution, 3D reconstitutions, surface or volume rendering, gamma adjustments, etc.).

8) References: There is no limit to the number of references cited in a manuscript. References should be cited parenthetically in the text by author and year of publication. Abbreviate the names of journals according to PubMed.

9) Supplemental materials: Articles may have up to 5 supplemental figures and 10 videos. You currently exceed this limit but, in this case, we will be able to give you the extra space. Please also note that tables, like figures, should be provided as individual, editable files. A summary of all supplemental material should appear at the end of the Materials and methods section. Please include one brief sentence per item.

10) Video legends: Should describe what is being shown, the cell type or tissue being viewed (including relevant cell treatments, concentration and duration, or transfection), the imaging method (e.g., time-lapse epifluorescence microscopy), what each color represents, how often frames were collected, the frames/second display rate, and the number of any figure that has related video stills or images.

11) eTOC summary: A ~40-50 word summary that describes the context and significance of the findings for a general readership should be included on the title page. The statement should be written in the present tense and refer to the work in the third person. It should begin with "First author name(s) et al..." to match our preferred style.

13) A separate author contribution section is required following the Acknowledgments in all research manuscripts. All authors should be mentioned and designated by their first and middle initials and full surnames. We encourage use of the CRediT nomenclature (<https://casrai.org/credit/>).

14) ORCID IDs: ORCID IDs are unique identifiers allowing researchers to create a record of their various scholarly contributions in a single place. Please note that ORCID IDs are required for all authors. At resubmission of your final files, please be sure to provide your ORCID ID and those of all co-authors.

15) JCB requires authors to submit Source Data used to generate figures containing gels and Western blots with all revised manuscripts. This Source Data consists of fully uncropped and unprocessed images for each gel/blot displayed in the main and supplemental figures. For assays performed using capillary electrophoresis and/or immunoassay-based detection, authors should instead provide the electropherogram graph(s) for each experiment, plotting fluorescence/chemiluminescence intensity vs. molecular weight/size. Since your paper includes cropped gel and/or blot images, please be sure to provide one Source Data file for each figure gels, blots, and/or capillary electrophoresis assays along with your revised manuscript files. File names for Source Data figures should be alphanumeric without any spaces or special characters (i.e., SourceDataF#, where F# refers to the associated main figure number or SourceDataFS# for those associated with Supplementary figures). For traditional gels and blots, the lanes of the gels/blots should be labeled as they are in the associated figure, the place where cropping was applied should be marked (with a box), and molecular weight/size standards should be labeled wherever possible. For capillary electrophoresis assays, each trace in the graph should be color-coded and labeled to indicate which protein, gene, or sample is being measured (please try to avoid red/green combinations to accommodate our color-blind readers).

Source Data files will be directly linked to specific figures in the published article. Source Data Figures should be provided as individual PDF files (one file per figure). Authors should endeavor to retain a minimum resolution of 300 dpi or pixels per inch. Please review our instructions for export from Photoshop, Illustrator, and PowerPoint here: <https://rupress.org/jcb/pages/submission-guidelines#revised>.

16) Journal of Cell Biology now requires a data availability statement for all research article submissions. These statements will be published in the article directly above the Acknowledgments. The statement should address all data underlying the research presented in the manuscript. Please visit the JCB instructions for authors for guidelines and examples of statements at

(<https://rupress.org/jcb/pages/editorial-policies#data-availability-statement>).

B. FINAL FILES:

-- A response to reviewers document addressing the remaining comments point by point. Please also highlight all changes in the text of the manuscript.

Thank you for your attention to these final processing requirements. Please revise and format the manuscript and upload materials within 7 days. If you need an extension for whatever reason, please let us know and we can work with you to determine a suitable revision period.

Thank you for this interesting contribution, we look forward to publishing your paper in Journal of Cell Biology.

Sincerely,

Karin Reinisch, PhD
Monitoring Editor
Journal of Cell Biology

Dan Simon, PhD
Scientific Editor
Journal of Cell Biology

Reviewer #1 (Comments to the Authors (Required)):

This is a revised version of previous manuscript. The authors have addressed my comments properly and the revised manuscript is substantially improved.

Reviewer #2 (Comments to the Authors (Required)):

The manuscript is improved with additional discussion and new figures highlighting the novelty of the mature α -HL structure presented here compared to the crystal structures and detergent-derived structures. However, the mature pore structure presented here appears almost identical to the previous liposome-derived structure published by the same group. Even the shift at residues S262 and G180 by 2.4 Å and ~2 Å respectively was shown in the Chatterjee et al., Nat Comm, 2025, from same

group (Fig S7b). Also, it is not clear what is meant by "... α -HL could not form an entire pore in the presence of sphingomyelin-containing PC membrane, and liposomes containing shorter chain length of PCs (Chatterjee et al., 2025)" (lines 448-450), considering in Chatterjee et al., 2025, they state "...we demonstrated the near-atomic resolution cryo-EM structures of mature pore and intermediate pre-pore conformations of α -HL in the presence of a lipidic environment."

The liposome-derived α -HL structure (Chatterjee et al. 2025) should be mentioned in the introduction. Lines 99-113 discuss previous pre-pore structures but the liposome-derived pre-pore structure (Chatterjee et al, 2025), is not mentioned. In the rebuttal, it is stated that the pre-pore state IV compared to the pre-pore state from liposomes (Chatterjee et al, 2025) are almost identical with an RMSD of 0.17. Although lines 274-275 state "However, the RMSD was very low when compared to the liposome-derived cryo-EM structure (Figure 3B)" the following sentences should also include the differences observed between the previous crystal and detergent structures with the liposome-derived structure.

For non-experts it would be beneficial to include a chart listing all the α -HL pore structures (pdbs) and corresponding methods, i.e. cryo-EM, x-ray, lipid, detergent.

The figure legends are much improved.

For Fig 3D, the insert zoomed view does not match boxed region in the lower mag image on the left making it hard to orient the shifts in residues relative to the monomer and lipid head groups. Since the shift around G180 and S262 was previously shown in Chatterjee et al, 2025, this figure could be supplementary.

The significant finding in this manuscript is the observation of intermediate species of α -HL pore formation in a native membrane environment, from 5-mer to a complete octameric pore.

We would like to thank the Editor for reviewing our manuscript and for sharing their suggestions to improve its quality. Additionally, we would also like to thank both reviewers for their time and effort in thoroughly reviewing our manuscript. We are grateful for their constructive suggestions and comments on our manuscript, which helped us to improve it.

1) Text limits: Character count for Articles is < 40,000, not including spaces. Count includes title page, abstract, introduction, results, discussion, and acknowledgments. Count does not include materials and methods, figure legends, references, tables, or supplemental legends.

Response: The character count of this article, including title page, abstract, introduction, results, discussion, and acknowledgment, is below 40,000, not including spaces.

2) Figure formatting: Articles may have up to 10 main text figures. Scale bars must be present on all microscopy images, including inset magnifications. Molecular weight or nucleic acid size markers must be included on all gel electrophoresis. Please add scale bars to figures 1E(iv) and magnifications in 2A & S3B.

Response: Thank you for bringing this to our attention. We have added a scale bar to Figure 1E(iv) and included the magnifications of the micrographs in Figures 2A & S3B in the figure legends.

Also, please avoid pairing red and green for images and graphs to ensure legibility for color-blind readers. If red and green are paired for images, please ensure that the particular red and green hues used in micrographs are distinctive with any of the colorblind types. If not, please modify colors accordingly or provide separate images of the individual channels.

Response: Thank you for your suggestions. We have changed the color in the bar graph in Figure 2D to avoid this issue.

3) Statistical analysis: Error bars on graphic representations of numerical data must be clearly described in the figure legend. The number of independent data points (n) represented in a graph must be indicated in the legend. Please indicate whether 'n' refers to technical or biological replicates (i.e. number of analyzed cells, samples or animals, number of independent experiments). If independent experiments with multiple biological replicates have been performed, we recommend using distribution-reproducibility SuperPlots (please see Lord et al., JCB 2020) to better display the distribution of the entire dataset, and report statistics (such as means, error bars, and P values) that address the reproducibility of the findings.

Response: We have mentioned “Data are shown as mean \pm SD of triplicate measurements (n = 3, biological triplicates)” in the figure legends of the bar graphs shown for cell death assays.

Statistical methods should be explained in full in the materials and methods. For figures presenting pooled data the statistical measure should be defined in the figure legends. Please

also be sure to indicate the statistical tests used in each of your experiments (both in the figure legend itself and in a separate methods section) as well as the parameters of the test (for example, if you ran a t-test, please indicate if it was one- or two-sided, etc.). Also, if you used parametric tests, please indicate if the data distribution was tested for normality (and if so, how). If not, you must state something to the effect that "Data distribution was assumed to be normal but this was not formally tested."

Response: We have not done any statistical analysis.

4) Title: While your current title will be appreciated by the specialists, we do not feel that it will be accessible to a broader cell biology audience. Therefore to convey the major advance more clearly, we suggest the following title: "Stepwise assembly of α -Hemolysin from a 5-mer to a complete octameric pore in native erythrocytes"

Response: Thank you for your suggestions. We really appreciate your suggestions. However, we have searched published papers and realized that no one is using 5-mer or octamer in the title. Also, our study didn't show that only transformations from 5-mer to octamer. It is basically assembled monomer to heptamer and thus, I think it is not a good idea to use 5-mer in the title. We have modified the title according to your suggestions. Although we did not include the exact geometrical shape in the title, we used a generic one for better clarity to the readers. The title looks like "**Stepwise assembly of α -hemolysin from intermediates to the mature pore in native erythrocytes**". I hope you are fine with this.

5) Materials and methods: Should be comprehensive and not simply reference a previous publication for details on how an experiment was performed. Please provide full descriptions (at least in brief) in the text for readers who may not have access to referenced manuscripts. The text should not refer to methods "...as previously described." Please make sure to provide a full description of the SDS-PAGE and immunoblotting procedures including the type of gel & membrane, blocking reagents, antibodies, and acquisition and quantification methods. Centrifugation speeds should be given in rcf/xg units and not rpm.

Response: Thank you for your suggestions. We have provided a detailed method for every experiment. The centrifugation speeds are now provided in xg.

6) For all cell lines, vectors, strains, constructs/cDNAs, etc. - all genetic material: please include database / vendor ID (e.g. Addgene, ATCC, etc.) or if unavailable, please briefly describe their basic genetic features, even if described in other published work or gifted to you by other investigators (and provide references where appropriate). Please be sure to provide the sequences for all of your oligos: primers, si/shRNA, RNAi, gRNAs, etc. in the materials and methods. You must also indicate in the methods the source, species, and catalog numbers/vendor identifiers (where appropriate) for all of your antibodies, including secondary. If antibodies are not commercial, please add a reference citation if possible.

Response: We have provided details of the cell line used in the method section and primer details in Supplementary Table 3.

7) Microscope image acquisition: The following information must be provided about the acquisition and processing of images:

- Make and model of microscope
- Type, magnification, and numerical aperture of the objective lenses
- Temperature
- Imaging medium
- Fluorochromes
- Camera make and model
- Acquisition software
- Any software used for image processing subsequent to data acquisition. Please include details and types of operations involved (e.g., type of deconvolution, 3D reconstitutions, surface or volume rendering, gamma adjustments, etc.).

Response: We have mentioned the microscopy details in the method section.

8) References: There is no limit to the number of references cited in a manuscript. References should be cited parenthetically in the text by author and year of publication. Abbreviate the names of journals according to PubMed.

Response: We have followed a similar pattern for referencing used by other articles published in JCB journal.

9) Supplemental materials: Articles may have up to 5 supplemental figures and 10 videos. You currently exceed this limit but, in this case, we will be able to give you the extra space. Please also note that tables, like figures, should be provided as individual, editable files. A summary of all supplemental material should appear at the end of the Materials and methods section. Please include one brief sentence per item.

Response: Thank you for your suggestions. We have provided a summary of all supplemental material at the end of the Materials and Methods section.

10) Video legends: Should describe what is being shown, the cell type or tissue being viewed (including relevant cell treatments, concentration and duration, or transfection), the imaging method (e.g., time-lapse epifluorescence microscopy), what each color represents, how often frames were collected, the frames/second display rate, and the number of any figure that has related video stills or images.

Response: As per your suggestions, we have provided all the details related to the videos provided in the manuscript.

11) eTOC summary: A ~40-50 word summary that describes the context and significance of the findings for a general readership should be included on the title page. The statement should be written in the present tense and refer to the work in the third person. It should begin with "First author name(s) et al..." to match our preferred style.

Response: Chatterjee et al. reveal that structural snapshots of different conformations of α -HL toxin from *S. aureus*, such as arc-like intermediates, heptameric pre-pores, and pore, and

octameric species, identified from the pre-hemolytic and post-hemolytic stages, offer step-by-step oligomerization of α -HL during erythrocyte membrane lysis.

Response: The authors declare no competing financial interests, as mentioned in the main manuscript.

13) A separate author contribution section is required following the Acknowledgments in all research manuscripts. All authors should be mentioned and designated by their first and middle initials and full surnames. We encourage use of the CRediT nomenclature (<https://casrai.org/credit/>).

Response: We have provided an author contribution section following the acknowledgments in the main manuscript.

14) ORCID IDs: ORCID IDs are unique identifiers allowing researchers to create a record of their various scholarly contributions in a single place. Please note that ORCID IDs are required for all authors. At resubmission of your final files, please be sure to provide your ORCID ID and those of all co-authors.

Response: We have added ORCID IDs for all the authors.

15) JCB requires authors to submit Source Data used to generate figures containing gels and Western blots with all revised manuscripts. This Source Data consists of fully uncropped and unprocessed images for each gel/blot displayed in the main and supplemental figures. For assays performed using capillary electrophoresis and/or immunoassay-based detection, authors should instead provide the electropherogram graph(s) for each experiment, plotting fluorescence/chemiluminescence intensity vs. molecular weight/size. Since your paper includes cropped gel and/or blot images, please be sure to provide one Source Data file for each figure gels, blots, and/or capillary electrophoresis assays along with your revised manuscript files. File names for Source Data figures should be alphanumeric without any spaces or special characters (i.e., SourceDataF#, where F# refers to the associated main figure number or SourceDataFS# for those associated with Supplementary figures). For traditional gels and blots, the lanes of the gels/blots should be labeled as they are in the associated figure, the place where cropping was applied should be marked (with a box), and molecular weight/size standards should be labeled wherever possible. For capillary electrophoresis assays, each trace in the graph should be color-coded and labeled to indicate which protein, gene, or sample is being measured (please try to avoid red/green combinations to accommodate our color-blind readers).

Source Data files will be directly linked to specific figures in the published article. Source

Data Figures should be provided as individual PDF files (one file per figure). Authors should endeavor to retain a minimum resolution of 300 dpi or pixels per inch. Please review our instructions for export from Photoshop, Illustrator, and PowerPoint here: <https://rupress.org/jcb/pages/submission-guidelines#revised>.

Response: We have provided Source data files based on your suggestions.

16) Journal of Cell Biology now requires a data availability statement for all research article submissions. These statements will be published in the article directly above the Acknowledgments. The statement should address all data underlying the research presented in the manuscript. Please visit the JCB instructions for authors for guidelines and examples of statements at (<https://rupress.org/jcb/pages/editorial-policies#data-availability-statement>).

Response: Thank you for your suggestions. We have provided a data availability statement above Acknowledgments.

B. FINAL FILES:

-- A response to reviewers document addressing the remaining comments point by point. Please also highlight all changes in the text of the manuscript.

****It is JCB policy that if requested, original data images must be made available to the editors. Failure to provide original images upon request will result in unavoidable delays in publication. Please ensure that you have access to all original data images prior to final submission.****

****The license to publish form must be signed before your manuscript can be sent to production. A link to the electronic license to publish form will be sent to the corresponding author only. Please take a moment to check your funder requirements before choosing the appropriate license.****

Additionally, JCB encourages authors to submit a short video summary of their work. These videos are intended to convey the main messages of the study to a non-specialist, scientific

audience. Think of them as an extended version of your abstract, or a short poster presentation. We encourage first authors to present the results to increase their visibility. The videos will be shared on social media to promote your work. For more detailed guidelines and tips on preparing your video, please visit <https://rupress.org/jcb/pages/submission-guidelines#videoSummaries>.

Thank you for your attention to these final processing requirements. Please revise and format the manuscript and upload materials within 7 days. If you need an extension for whatever reason, please let us know and we can work with you to determine a suitable revision period.

Thank you for this interesting contribution, we look forward to publishing your paper in Journal of Cell Biology.

Reviewer #1 (Comments to the Authors (Required)):

This is a revised version of previous manuscript. The authors have addressed my comments properly and the revised manuscript is substantially improved.

Reviewer #2 (Comments to the Authors (Required)):

Comment: The manuscript is improved with additional discussion and new figures highlighting the novelty of the mature α -HL structure presented here compared to the crystal structures and detergent-derived structures. However, the mature pore structure presented here appears almost identical to the previous liposome-derived structure published by the same group. Even the shift at residues S262 and G180 by 2.4 Å and ~2 Å respectively was shown in the Chatterjee et al., Nat Comm, 2025, from same group (Fig S7b).

Response: Thank you for your positive comments and suggestions. Previously, the α -HL structures were solved in the presence of detergent, or detergent-exchanged lipid, or with artificial phosphatidyl-choline containing liposomes. However, the α -HL pore conformation solved in the presence of liposomes is identical to the pore structure in the presence of erythrocytes. As per your suggestion, we have shifted these minimal structural changes from Figure 3D to Supplementary Figure S3E. However, this study will be significant due to a broader audience, as mentioned below.

Our lab resolved the structure of VCC toxin in the presence of liposomes (<https://doi.org/10.1083/jcb.202102035>), which inspired many groups to solve the structures of lipid-binding proteins, including pore-forming toxins (<https://doi.org/10.1038/s41467-023-41579-x>, <https://doi.org/10.1016/j.str.2023.03.009>, <https://doi.org/10.1016/j.jsb.2024.108116>). Similarly, our current study is one of the first studies where we resolved various conformational variabilities of hemolysin in the presence of RBC. We are able to demonstrate that, without doing cryo-electron tomography, we can employ single particle analysis and characterize these small pore-forming toxins in the

cellular environment. This current study will motivate researchers to move forward with the structure determination in the real cellular environment, where they will be able to capture more intermediate conformations as compared to artificial lipidic conditions. Additionally, changes in the plasma membrane during the toxin attack are demonstrated using high-resolution imaging, which highlights the importance of toxin-plasma membrane interaction in native conditions.

Comment: Also, it is not clear what is meant by "... α -HL could not form an entire pore in the presence of sphingomyelin-containing PC membrane, and liposomes containing shorter chain length of PCs (Chatterjee et al., 2025)" (lines 448-450), considering in Chatterjee et al., 2025, they state "...we demonstrated the near-atomic resolution cryo-EM structures of mature pore and intermediate pre-pore conformations of α -HL in the presence of a lipidic environment."

Response: Thank you for bringing this concern to our attention. We have revised the sentence in the discussion section of the main manuscript for better clarity and readability. The sentence read like, in lines 439-442, and pages 18-19, is "However, structural analysis of α -HL in the presence of sphingomyelin-containing PC-containing liposomes revealed heptameric pre-pore species. Whereas the PC-containing liposomes without sphingomyelin generated heptameric pore conformation". This happened because sphingomyelin makes the lipid more rigid, and PFT is unable to punch a hole and form pre-pore conformations; whereas PC-containing liposomes are more flexible and easier to punch holes and form a complete pore structure.

Comment: The liposome-derived α -HL structure (Chatterjee et al. 2025) should be mentioned in the introduction. Lines 99-113 discuss previous pre-pore structures but the liposome-derived pre-pore structure (Chatterjee et al, 2025), is not mentioned.

Response: Thank you for pointing that out. We have included this line in the introduction section of the main manuscript on page 5, and lines 111-113 "Similarly, high-resolution pore and pre-pore conformations of heptameric α -HL were resolved in the presence of PC-containing liposomes".

Comment: In the rebuttal, it is stated that the pre-pore state IV compared to the pre-pore state from liposomes (Chatterjee et al, 2025) are almost identical with an RMSD of 0.17. Although lines 274-275 state "However, the RMSD was very low when compared to the liposome-derived cryo-EM structure (Figure 3B)" the following sentences should also include the differences observed between the previous crystal and detergent structures with the liposome-derived structure.

Response: Thank you for your query. We have mentioned the RMSD values between the pore structure in the presence of RBCs with MPD solubilized pore (PDB:3ANZ), pore crystal structure with PC-head group (PDB: 6U49), 10:0 PC-liposome derived pore structure (PDB:9KRF), and detergent-solubilized pore structure (PDB: 7AHL) in page 12, lines 261-265. Therefore, the sentence in lines 270-271, "However, the RMSD was very low when compared to the liposome-derived cryo-EM structure (Figure 3B)," appears redundant and confusing to the readers. Thus, we have removed this sentence from the main manuscript text.

Comment: For non-experts it would be beneficial to include a chart listing all the α -HL pore structures (pdbs) and corresponding methods, i.e. cryo-EM, x-ray, lipid, detergent.

Response: Thank you for your insights. In this manuscript, we have mentioned all the previously solved structures and provided PDB IDs for all in Figure 3. However, we have noticed that the requested kinds of charts are generally provided in review articles (<https://doi.org/10.1099/mic.0.001154>, <https://doi.org/10.1038/nrmicro.2015.3>, <https://doi.org/10.1007/s00232-025-00344-5>) and not in research articles. Therefore, we are a little bit sceptical about including this chart in the main text. In the future, we plan to write a review on the structural analysis of α -HL structures under various conditions. We will provide a chart listing all the structures of α -HL in the review.

Comment: The figure legends are much improved.

Response: Thank you for your comments.

Comment: For Fig 3D, the insert zoomed view does not match boxed region in the lower mag image on the left making it hard to orient the shifts in residues relative to the monomer and lipid head groups. Since the shift around G180 and S262 was previously shown in Chatterjee et al, 2025, this figure could be supplementary.

Response: Thank you for your suggestion. In the revised figures, we have moved the comparison to a supplementary figure (Figure S3E).

Figure S3E: Comparison of pore structure solved in the presence of RBCs (cyan) with pore structure solved in the presence of detergent (PDB ID: 7AHL, pink). The shifts in S262 and G180 residues in the rim domain are 2.4 Å and ~2 Å, respectively.

Comment: The significant finding in this manuscript is the observation of intermediate species of α -HL pore formation in a native membrane environment, from 5-mer to a complete octameric pore.

Response: We appreciate your positive comments.